# RaLEs: a Benchmark for Radiology Language Evaluations

**Juan M Zambrano Chaves**
Stanford University
jmz@stanford.edu

**Nandita Bhaskhar**
Stanford University

**Maayane Attias**
Stanford University

**Jean-Benoit Delbrouck**
Stanford University

**Daniel L. Rubin**
Stanford University

**Andreas Loening**
Stanford University

**Curtis Langlotz**
Stanford University

**Akshay S. Chaudhari**
Stanford University
akshaysc@stanford.edu

## Abstract

The radiology report is the main form of communication between radiologists and other clinicians. Prior work in natural language processing in radiology reports has shown the value of developing methods tailored for individual tasks such as identifying reports with critical results or disease detection. Meanwhile, English and biomedical natural language understanding benchmarks such as the General Language Understanding and Evaluation as well as Biomedical Language Understanding and Reasoning Benchmark have motivated the development of models that can be easily adapted to address many tasks in those domains. Here, we characterize the radiology report as a distinct domain and introduce RaLEs, the Radiology Language Evaluations, as a benchmark for natural language understanding and generation in radiology. RaLEs is comprised of six natural language understanding and generation evaluations including the extraction of anatomical and disease entities and their relations, procedure selection, and report summarization. We characterize the performance of models designed for the general, biomedical, clinical and radiology domains across these tasks. We find that advances in the general and biomedical domains do not necessarily translate to radiology, and that certain more advanced models from the general domain can perform comparably to smaller clinical-specific models. The limited performance of existing pre-trained models on RaLEs highlights the opportunity to improve domain-specific self-supervised models for natural language processing in radiology. We propose RaLEs as a benchmark to promote and track the development of such domain-specific radiology language models. RaLEs is available at https://github.com/StanfordMIMI/RaLEs.

## 1 Introduction

Radiology reports convey a radiologist's interpretation of a medical image. The reports have characteristic content and structure that differentiate them from other types of text. Natural language processing of radiology reports can aid research efforts and ultimately lead to improved quality of care. However, many recent approaches focus on solving single radiology tasks, often report performance only on private datasets, fail to compare proposed methods with relevant baselines, and do not publish code or models [5]. Further, reports are typically not publicly available due to patient privacy concerns. The development of private, single-task models limits the measurement of progress in NLP for radiology broadly.

37th Conference on Neural Information Processing Systems (NeurIPS 2023) Track on Datasets and Benchmarks.

Model benchmarking in other domains, such as general English or biomedical text, has enabled thorough comparisons of existing methods across tasks that evaluate model performance and alignment with human evaluation [53; 20; 35]. This has promoted the development of state of the art models that can be adapted to address multiple tasks. Radiology reports are excluded from existing biomedical or clinical benchmarks, which often feature evaluations on datasets that do not reflect real-world clinical use cases. In contrast, 75% of current current FDA-approved AI applications target radiology ([18]), with a potential for impact on over 700 million studies (and associated reports) annually [39]. However, it is unclear how advances in other domains translate to the unique domain of radiology language and reports.

To address the aforementioned challenges, in this work we develop RaLEs, a benchmark for evaluations of natural language understanding (NLU) and generation (NLG) in the radiology domain. Our main contributions are:

1. Curating a set of 6 datasets across 4 tasks, all of which are publicly available. Among these datasets, 1 is newly created and introduced in this work (procedure selection), and 1 is newly de-identified and released into the public domain (Stanza radiology named entity recognition).

2. We benchmark and report RaLEs multi-metric performance on 16 models from the general, biomedical, clinical and radiology domains. We find that on average, clinical and radiology-specific models outperform general and biomedical models by 1.5 and 0.5% on preferred metrics.

3. We consolidate progress by developing RaLEs NLU and NLG scores, and release code for dataset standardization, model fine-tuning and evaluation to spur future benchmarking.

## 2 Related work

### 2.1 Radiology natural language processing

Prior work has reviewed the status of NLP in radiology. In a systematic review, Casey et al. [5] identified that <20% of previously published radiology NLP methods used deep learning, with the majority relying on other machine learning or rule-based systems. A significant portion of these studies primarily concentrated on tasks like information extraction (accounting for 45%) and classification (making up 50%). Regarding reproducibility, only 14 and 15 of the 164 studies reviewed made their data and code available, respectively. Fewer than 20% of studied compared proposed methods with alternate approaches. In a separate study, Pons et al. [44] reported that 20 of 67 studies reported operational use, the majority of which were not intended for integration into a clinical workflow.

Most existing applications of NLP in radiology can be framed as NLU or NLG tasks. These include information extraction (named entity recognition and/or relation extraction), text classification, and text summarization [40; 38]. These methods can enable applications such as extracting labels from reports to annotate images [26; 48], automatic image protocol selection, defining patient cohorts, monitoring appropriate use and clinical follow-up of medical images, and summarizing prior imaging studies [40; 38].

### 2.2 Benchmarks in other domains

Benchmarks that systematically compare the performance of existing models in the general English domain, such as GLUE or SuperGLUE [53; 52], have enabled comparisons of models across a variety of NLU tasks, promoting the development of pre-trained models that can be adapted to a variety of tasks using transfer learning or other adaptation methods. Other benchmarks have been proposed for biomedical language understanding and reasoning, such as the BLURB Benchmark [20], which contains 13 datasets and 6 tasks, focusing on the performance of systems across scientific biomedical text. The Biomedical Language Understanding Evaluation (BLUE) benchmark [43], includes 6 biomedical scientific text datasets and 4 clinical datasets evaluating sentence similarity, named entity recognition, relation extraction, document classification and inference. While comprehensive, the BLUE benchmark does not include any radiology text-based tasks. This highlights a broader issue: no existing benchmarks broadly compare the performance of models in a curated set of radiology tasks, which makes it challenging to quantify the efficacy of general-purpose language models on domain-specific radiology tasks.

# 3 Radiology reports as a domain

Three key characteristics define the domain of radiology reports: a) content, b) context and c) closed source. In terms of content, radiology reports contain a limited specialized vocabulary, often existing only in the context of images. For example, words like *pneumothorax*, *cardiomegaly*, and *radiodensity*, referring to air in the space surrounding the lungs, an enlarged heart, and opacity to X-rays, are frequently found in radiology reports. Such words are contained in Radiology Lexicon[1], a comprehensive set of radiology terms that contains approximately 30,000 entries. An additional aspect of content involves structure. Radiology reports typically follow a document structure comprised of a header containing patient history and exam-related information, followed by mentions of relevant comparison studies, details about the imaging technique utilized, a detailed description of image findings, and an impression that summarizes findings contextualized to the patient's condition [30; 23]. Sentences within reports are typically short, declarative and factual, and are written in the present tense. The content (unique vocabulary, format, and narrative style) of radiology reports is an important feature of this specialized domain.

Another aspect that characterizes this domain is the context surrounding radiology reports. Reports only exist in the presence of an accompanying medical image. Typically they exist within the context of electronic medical records, which are collections of documents, images and other signals that describe a patient's medical history. The content of these reports is embedded within the current radiology and medical knowledge.

Finally, due to existing concerns and regulations that protect patient health information, the vast majority of radiology reports exist within private data warehouses, often requiring institutional review board approval for access. The largest publicly available collections of reports, such as MIMIC-III, MIMIC-CXR, PadChest, and Open-i datasets [29; 28; 4; 15], have undergone de-identification and are typically made available subject to agreement to terms of data use protecting patient privacy.

# 4 RaLEs: Radiology Language Evaluations

The following sections outline RaLEs. The tasks and datasets were selected to reflect real-world use cases, ensuring they are both challenging and achievable. We outline multiple metrics of success for well-rounded evaluation for each dataset-task pair, as well as differentiate the performance of the model in data from institutions unseen during training where available.

## 4.1 Tasks and datasets

Table 1: Overview of datasets and tasks in RaLEs.

| Category | Dataset | Task | # Train/Dev/Test | Anatomy/Modality | # Sources | New |
|---|---|---|---|---|---|---|
| NLU | RadGraph [27] | RE | 425 / 75 / 100 | Chest / XR | 2 | |
| | RadGraph [27] | NER | 425 / 75 / 100 | Chest / XR | 2 | |
| | RadSpRL [11] | RE | 848 / 105 / 106 | Chest / XR | 1 | |
| | Stanza Radiology [58] | NER | 2,461 / 200 / 295 | Chest / CT | 3 | ✔ |
| | CT Procedure Selection | Clf. | 58,091 / 19,364 / 19,364 | Varied / CT | 1 | ✔ |
| NLG | MEDIQA 2021[3] | Summ. | 91,544 / 4,000 / 600 | Chest / XR | 2 | |
| | BioNLP 2023[14] | Summ. | 59,320 / 7,413 / 13,057 | Varied / CT, MR | 1 | |

XR=X-ray, CT=Computed Tomography, MR=Magnetic Resonance Imaging, NLU=Natural Language Understanding, NLG=Natural Language Generation, New= newly released with this work, Varied=Abdomen, Pelvis, Neck, Spine, Head, etc., see A and B

Table 1 summarizes tasks and datasets within RaLEs. We include four tasks as part of the initial RaLEs: named entity recognition (NER), relation extraction (RE), document classification and document summarization. The datasets selected for each task are described as follows.

**RadGraph** [27] consists of 600 manually annotated chest x-ray reports from MIMIC-CXR and CheXpert datasets [28; 26]. A board-certified radiologist labeled the reports with named entities, consisting of *Observation - definitely present*, *Observation - definitely absent*, *Observation - uncertain*, and *Anatomy*, reflecting key entities in reports corresponding to observations (which may be negated or hedged) as well as anatomical structures. Pairs of entities may be related by one of three types: *suggestive of*, *located at*, and *modify*. We use this dataset to evaluate models on NER and RE. Test

---

[1]https://www.radlex.org

set evaluations are carried out separately on MIMIC-CXR and CheXpert, reflecting in and out of distribution performance, respectively.

**RadSpRL** [11] consists of 2000 manually annotated chest x-ray reports from the Open-i dataset [15]. A medical librarian and an MD annotated entities and relations that represent spatial relations. Spans of text were annotated as a relationship between a *Spatial indicator* with another span of text consisting of one of four spatial roles: *Trajector*, *Landmark*, *Hedge*, and *Diagnosis*. We evaluate models on RE with this dataset. We use only documents with labeled relations for training and evaluation. Though prior performance is reported on cross-validation sets, we create a fixed train/dev/test split on the report level to limit excessive compute requirements of hyperparameter exploration across each split for each model.

**Stanza Radiology** [58] consists of 150 manually annotated chest computed tomography reports from three hospitals. Two radiologists annotated spans of text in the reports with five entity types: *Anatomy*, *Observation*, *Anatomy modifier*, *Observation modifier*, and *Uncertainty*. This dataset is used for NER evaluations. As part of RaLEs, we have deidentified these reports using a publicly available hidden-in-plain-sight de-identification algorithm [7], and release this previously private dataset to the public.

**MIMIC-III procedure selection** is a newly created dataset, released alongside this work, that consists of 96,819 documents extracted from individual computed tomography reports from the MIMIC-III dataset [29]. The reason for exam and procedure title were extracted from each report using regular expressions. The task consists of appropriately classifying reason for exam documents into one of 46 normalized procedure titles. The procedure titles were normalized to a standardized vocabulary [50] by manually mapping a set of extracted procedure titles to the vocabulary. Normalization was carried out by an MD and a board-certified radiologist. Additional details of the curation of this dataset are in A. This evaluation was developed to simulate the selection of a procedure given a clinician provided reason for exam, a task that often requires expert human oversight in current practice.

**MEDIQA 2021 report summarization** [3] consists of 96,144 chest X-ray reports with extracted *Findings* and *Impression* sections. The task is summarizing the *Findings* section of reports, using the *Impression* as ground truth. Models are trained using reports from one dataset (MIMIC-CXR) and validated using reports from MIMIC-CXR and the Open-I dataset (from Indiana). The test evaluation is carried out on reports from an institution seen during validation (Indiana), as well as an institution present only in the test set (Stanford), which aims to measure out-of-domain generalization.

**BioNLP 2023 report summarization** [14] consists of 79,790 multi-modal reports [8] extracted from the MIMIC-III dataset [29] that are separated into *Findings* and *Impression* sections. The task is to create a summary in the same fashion as MEDIQA 2021. The reports are of computed tomography and magnetic resonance imaging examinations, with head, chest, abdomen, spine and sinuses present as different anatomies (Details in B. Test evaluations include anatomies/modalities unseen in training data.

## 4.2 Evaluation strategy

### 4.2.1 Models

Table 2: Masked language models evaluated. Number of parameters in millions.

| Domain | Model | # Params |
|---|---|---|
| English | BERT | {110, 340} |
| | RoBERTa | {125, 355} |
| | ELECTRA | {14,110,345} |
| | DeBERTa-v3 | {86, 304} |
| Biomedical | PubMedBERT | 110 |
| | BioLinkBERT | {110, 340} |
| Clinical | BioClinicalBERT | 110 |
| | GatorTron | 345 |
| Radiology | RadBERT$_1$ | 110 |
| | RadBERT$_2$ | 125 |

We evaluate various pre-trained masked language models from the general, biomedical, clinical and radiology domains. Table 2 lists the models evaluated. We refer the reader to the respective publication for details on the pretraining strategy including source of vocabulary, corpus and model-specific optimizations.

From the general English domain, where models are trained using sources such as Wikipedia and Google Books, we evaluate BERT [17] in its base and large configurations, RoBERTa [36] in its base and large configurations, ELECTRA [9] in its small, base and large configurations, and DeBERTa-v3 [24] in its base and large configuration.

From the biomedical domain, we examine models pre-trained on scientific text contained in PubMed: PubMedBERT-base [19], which pretrains a BERT-base model using biomedical scientific abstracts and full-text articles as a corpus, and BioLinkBERT-base and large [56], which uses a similar pretraining corpus but incorporates an additional pretraining objective consisting of identifying document links.

From the clinical domain, we evaluate BioClinicalBERT [2], a BioBERT [34] biomedical model (i.e., a BERT-base model continually pretrained on PubMed text), which is further continually pretrained on MIMIC-III clinical notes. In addition, we examine GatorTron-base[55], a MegaTronBERT model [47] pretrained using clinical notes from the University of Florida, PubMed text and Wikipedia articles (500GB of text).

Finally, we examine the performance of models developed specifically for the radiology domain, both named RadBERT by their creators. One model, which we refer to as $RadBERT_1$, corresponds to a BioBERT model continually pretrained on 4 million radiology reports from Stanford Health Care [6]. The second model, which we refer to as $RadBERT_2$, corresponds to a BioMed-RoBERTa (RoBERTA-base model continually pretrained on 2.7 million scientific papers [22] ) model further pretrained on 4.4 million radiology reports from various facilities of the U.S. Department of Veterans Affairs health system.

### 4.2.2  NLU evaluations

We fine-tune each model using all documents in training split and perform task-specific hyperparameter optimization as detailed in C. We select the best model according to the best run preferred metric on the validation set, as defined by each dataset. We report the performance of models on test sets using the best model for each model type/task. We employ different fine-tuning strategies for each task, described as follows.

For RE, datasets (RadGraph, RadSpRL), we use DyGIE++ [51], a multi-task NER and RE framework that learns a dynamic graph that models relationships between text spans. Span representations are obtained from a language model embedding. We use DyGIE++ as a NER extraction method for RadGraph models as we empirically observed improved performance compared to the Stanza NER approach which cannot leverage the relation labels. For document classification, a randomly initialized head of dimensions $(h, c)$ where $h$ corresponds to the hidden size and $c$ corresponds to the number of classes, is added as a final layer to classify the classification [CLS] token. For Stanza NER, a randomly initialized head of dimensions $(h, c)$ is added to classify each token. For words comprised of multiple sub-word tokens, the label is assigned to the first token.

Consistent with prior evaluations for existing datasets, the preferred metric for all evaluations is the micro-averaged F1 score. For the newly created procedure selection task, we select accuracy as the preferred metric as it more closely reflects success in clinical settings. We report a RaLEs NLU score as the average of the preferred metrics across the NLU datasets. In addition, we perform label-stratified sampling evaluations using 1% and 10% labels during training and validation to assess how dataset scaling across domain-specific and domain-agnostic models affects their performance. We also examine separability of representations generated by each model by keeping language model weights frozen and training only a linear probe, or only the graph layers in the case of DyGIE++ models. Furthermore, we evaluate models in metrics that may be relevant for deployment, focusing on model calibration and uncertainty. We include in RaLEs metrics measuring calibration and uncertainty, prediction quality, and information criteria. We detail the results for the document classification (procedure selection) task on these metrics in Appendix D. Model fine-tuning is performed in private infrastructure using a single NVIDIA TITAN RTX or RTX A6000 GPU. For each model, hyperparameter exploration takes on average 1 hour for DyGIE++ models, 1 hour for Stanza NER, and 3 hours for the procedure selection task (in total approximately 400 GPU hours).

### 4.2.3 NLG evaluations

For report summarization, we fine-tune each model to predict the *Impressions* section of a radiology report given the *Findings*, using the MEDIQA2021 and BioNLP 2023 datasets. Specifically, we construct an encoder-decoder model for sequence generation [46] using the same pretrained model as the encoder and the decoder, and newly initialize a cross-attention layer. All weights are updated during model fine-tuning. Due to increased computational cost of hyperparameter exploration for each model (10-20 hours per training run on an RTX A6000 GPU), we chose a representative model from each domain for study: ELECTRA$_{base}$, BioLinkBERT$_{base}$, GatorTron, and RadBERT$_2$. Further, we include our evaluations of two recently proposed report summarization-specific approaches: RadiologyGPT and RadAdapt [37; 49].

We evaluate the quality of generated summaries using commonly used lexical similarity metrics, ROUGE-2, ROUGE-L [31; 25], as well as recently proposed metrics to evaluate factual correctness of a report [57; 12]. For factual correctness in Chest X-ray report summarization (MEDIQA 2021), we examine the F1-CheXbert score, which measures the micro-averaged F1 score of 14 disease mentions extracted from a generated summary, using the original as ground truth [57]. In addition, we report the F1-RadGraph [12] (RG) metric, which compares the agreement of anatomy and observation entities and their relations extracted in the generated versus the original summary. We report the test-set results using the ROUGE, CheXbert and RG metrics, and summarize model performance in the RaLEs NLG score, which is the average of the ROUGE-L and RG metrics across both datasets.

## 5 Results and discussion

### 5.1 NLU

Table 3: Summary of results for NLU tasks.

| Model | RG$_{NER}$† | RG$_{NER}$‡ | RG$_{RE}$† | RG$_{RE}$‡ | RadSpRL | Stanza | Procedure | NLU Score |
|---|---|---|---|---|---|---|---|---|
| BERT$_{base}$ | 93.0 | 86.3 | 82.7 | 70.6 | 91.2 | 83.7 | 65.4 | 81.8 |
| BERT$_{large}$ | 92.6 | 88.5 | 82.0 | 71.4 | 85.3 | 85.4 | 64.9 | 81.4 |
| RoBERTa$_{base}$ | 92.4 | 89.7 | 81.3 | 70.2 | 89.6 | 81.5 | 64.2 | 81.3 |
| RoBERTa$_{large}$ | 92.7 | 89.7 | 83.0 | 73.1 | 82.1 | 83.0 | 64.9 | 81.2 |
| ELECTRA$_{small}$ | 92.6 | 89.7 | 82.0 | 70.7 | 88.1 | 73.1 | 61.5 | 79.7 |
| ELECTRA$_{base}$ | 93.2 | 85.9 | 83.2 | 71.9 | 89.9 | 85.2 | 64.4 | 82.0 |
| ELECTRA$_{large}$ | 93.0 | 86.0 | 82.3 | 71.0 | 87.5 | 84.9 | 64.2 | 81.3 |
| DeBERTa-V3$_{base}$ | 93.4 | 89.8 | **84.6** | 73.4 | 89.8 | 84.1 | 65.7 | 83.0 |
| DeBERTa-V3$_{large}$ | 93.1 | 89.9 | 83.8 | 73.4 | 89.4 | 85.1 | 64.2 | 82.7 |
| PubMedBERT | 92.1 | 86.8 | 82.9 | 71.7 | 88.9 | 85.0 | 65.9 | 81.9 |
| BioLinkBERT$_{base}$ | 93.2 | 90.6 | 83.6 | **75.1** | 91.0 | 83.7 | 65.7 | 83.3 |
| BioLinkBERT$_{large}$ | 93.2 | 90.3 | 82.6 | 72.4 | 89.6 | 84.6 | 65.7 | 82.6 |
| BioClinicalBERT | 93.7 | 90.4 | 82.0 | 72.8 | 91.9 | **85.6** | 65.4 | 83.1 |
| GatorTron | 93.5 | 89.8 | 82.9 | 74.6 | **92.0** | 84.3 | **66.7** | **83.4** |
| RadBERT$_1$ | 93.8 | 90.1 | 81.2 | 72.1 | 91.2 | 85.1 | 65.4 | 82.7 |
| RadBERT$_2$ | **94.0** | **90.7** | 81.9 | 73.4 | 91.2 | 85.5 | 65.8 | 83.2 |
| Prior SOTA§ | **94.0** | 90.5 | 82.3 | 72.5 | 85.6 | 84.8 | - | - |

†: MIMIC-CXR (in domain), ‡: CheXpert (out of domain), §: RadGraph [27], RadSpRL [11], Stanza [58]

Table 3 summarizes the results of the NLU evaluation. No single model outperforms all others across all tasks. In general, biomedical or clinical models outperform general domain models. Furthermore, most models perform similarly (within 2-3 percentage points of each other; standard deviation of average performance across models is 1).

#### 5.1.1 Impact of domain, pretraining objective and model size

Figure 1 shows the summarized performance of models across domain and size. Overall, general English models perform worse than their more specialized counterparts. Furthermore, models in the clinical/radiology domain outperform those from the biomedical and general domains. This is consistent with prior observations of benefits of using domain specific corpora during pretraining[19; 55; 54]. This trend is consistent across varying availability of labeled fine-tuning data (Figure 2).

Additionally, improved pretraining objectives typically lead to improvements in RaLEs NLU performance. For example, ELECTRA models (adversarially trained to predict replaced tokens) outperform models trained only with masked language modeling objectives (such as BERT). DeBERTa-V3, which in addition to replaced token detection enhances token representation with disentangled attention,

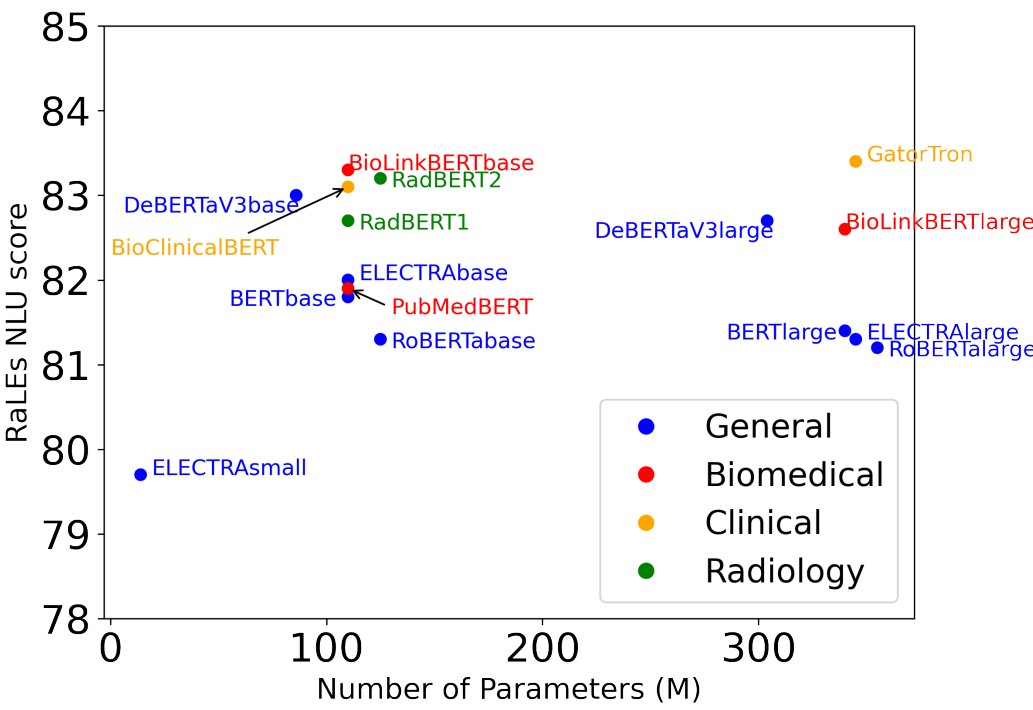

Figure 1: Model size and domain vs RaLEs NLU performance

obtains improvements in both in GLUE as well as in RaLEs NLU. Similarly, though BioLinkBERT and PubMedBERT share similar scientific pretraining corpora, the document relation prediction objective of BioLinkBERT leads to improved performance in both BLURB as well as RaLEs NLU.

However, improvements in performance in other benchmarks, such as GLUE or BLURB, do not directly translate to improvements in RaLEs NLU. This is illustrated in Figure 3 which shows that 10 percentage point improvements in GLUE or BLURB lead to a 1-2 percent improvement in RaLEs NLU performance. We hypothesize that the decrease in benefit stems from the domain characteristics of radiology that differentiate it from other text, as proposed in Section 3. Furhermore, we observe that existing radiology domain adaptation hurts alternate domain performance, exemplified by the relatively low BLURB score for RadBERT$_{1,2}$ models seen in Figure 3. Finally, we find that given a fixed architecture, an increase in parameter count ($O$(300M) vs $O$(100M)) does not lead to improved overall performance. This observations holds for both English and biomedical models, which are the only ones publicly available in different sizes. Similar results have been previously observed in other domains [20; 45], though the impact of base model size on domain adaptation is yet to be systematically studied.

## 5.2 NLG

Table 4 presents the results for the report summarization task, with prior results referenced for comparison. Similarly to NLU results, no model is consistently superior across all metrics. GatorTron, the model from the clinical domain, outperforms the other evaluated fine-tuned models slightly on lexical similarity (ROUGE-2 and ROUGE-L). Using these metrics as reference, the encoder-decoder framework as implemented here performs inferiorly to the best existing performing models. The best MEDIQA 2021 prior model [10], in addition to an abstractive-summarization-specific architecture, uses a domain adaptation module to improve performance on Indiana reports. However, as can be seen from our evaluation on fully held out Stanford reports in Appendix G, most of the benefits of such specialized approaches may not generalize to sites not seen during training. Further, our approach seems to favor factual correctness, with BioLinkBERT and GatorTron models having the best performance according to the CheXbert and RG metrics. RadAdapt [49], a label efficient adaptation of a clinical large language model, matches the overall NLG performance of our best fine-tuned model. As the RadAdapt authors recognize, however, it is unclear to what extent its

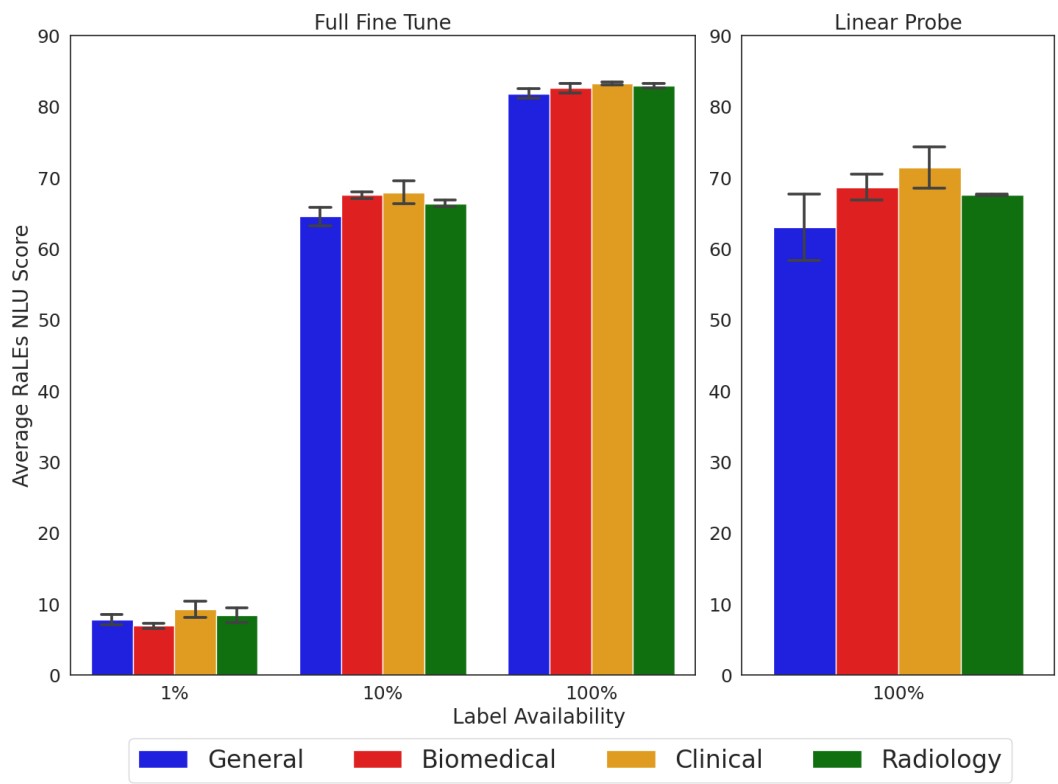

Figure 2: RaLEs NLU performance by model domain. Values are averaged across model domain, error bars are standard deviation. There are no statistically significant pairwise differences between model categories within the same label availability, as determined by Mann-Whitney U-tests with Bonferroni correction.

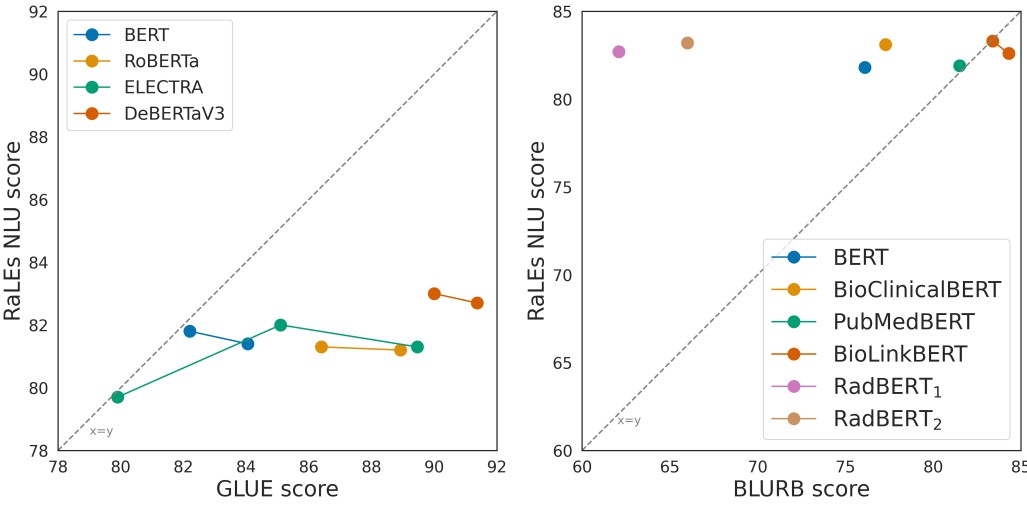

Figure 3: General (English, left) and biomedical (right) benchmark vs RaLEs NLU performance.

performance is affected by possible leakage of testing data during the base model pre-training stage. We aim to further evaluate other large language models in future versions of RaLEs.

Table 4: Summary of results for NLG tasks (abstractive report summarization).

| Model | MEDIQA 2021 | | | | BioNLP 2023 | | | NLG score |
|---|---|---|---|---|---|---|---|---|
| | **R-2** | **R-L** | **CheXbert** | **RG** | **R-2** | **R-L** | **RG** | |
| ELECTRA$_{base}$ | .238 | .381 | .710 | .378 | .156 | .274 | .229 | .316 |
| BioLinkBERT$_{base}$ | .245 | .388 | **.725** | .391 | .183 | .297 | .272 | .337 |
| GatorTron | .250 | .386 | .719 | **.406** | .189 | .303 | .283 | **.345** |
| RadBERT$_2$ | .237 | .382 | .709 | .381 | .184 | .300 | .271 | .334 |
| RadAdapt§ | **.253** | **.393** | - | .345 | **.212** | **.324** | **.342** | **.345** |
| RadiologyGPT§ | .074 | .148 | .601 | .135 | .127 | .209 | .242 | .184 |
| Prior SOTA† | **.436** | **.557** | .718 | - | - | - | - | - |
| Prior Baseline‡ | .264 | .389 | .610 | - | - | - | - | - |

R-2/L: ROUGE 2/L, RG: F1-RadGraph, †:MEDIQA[10], ‡:from [3], §:our evaluation of [49] and [37]

## 5.3 Ethical considerations

RaLEs provides a framework for benchmarking advances in radiology NLP. The current version of RaLEs uses the RadSpRL dataset, which has a CC BY 4.0 license. Datasets stemming from MIMIC, including RadGraph, have a PhysioNetCredentialed Health Data License 1.5.0 which prohibit commercial use, data sharing and patient or institution identification attempts. For the newly released Stanza NER dataset, the dataset will be accompanied by an analogous Research Use Agreement following institutional review board approval, which was obtained to access the reports. For all datasets we have followed the appropriate research use, have not attempted re-identification of individuals, and provide instructions for data access in the accompanying code. Demographic characteristics of individuals included in RadGraph [27] and MIMIC-III [29] datasets are described in their original publications. Demographic characteristics for the Stanza NER [58], MEDIQA 2021 [3] and Indiana [16] dataset (from which the RadSpRL dataset is derived) are not reported since the radiology reports have been de-identified and these characteristics are not otherwise available. We do not foresee any potential risks following appropriate use of RaLEs as a tool to measure progress in NLP research. We strongly discourage the use of models trained using our framework for clinical care or advice without adequately studying the performance and limitations on specific patient populations that reflect intended use.

## 5.4 Limitations

An important limitation of our analysis stems from the limited availability of publicly available radiology reports and datasets. This is consistent with prior observations that healthcare algorithms trained on US patient data rely on data from a handful of states [32]. While we release a new dataset from a new institution, future efforts should promote the availability of training and/or evaluation datasets from additional institutions, geographic locations, and languages. Aside from geographic diversity, we note that most publicly available reports and evaluations focus on chest X-ray reports which tend to be shorter and simpler than reports of other modalities and anatomies (see example lengths in Table 13). Our newly introduced procedure selection dataset expands the scope of current data by focusing on a different modality (computed tomography) across all anatomies. Finally, while we present extensive evaluations across models of different domains, we intend RaLEs to be a dynamic benchmark, with expansions across newer datasets, tasks, and model evaluations (including adding newer models and estimates of variance of performance).

## 6 Conclusion

Radiology reports are defined by their content, context and restricted access. RaLEs defines a benchmark for meassuring progress in NLP in the radiology domain, focusing on multifaceted evaluations that reflect real-world use cases in radiology research and/or practice. The results showed that no single model, including existing radiology-specific domain adapted models, outperforms others across all evaluations.

# 7 Acknowledgements

We thank Dave Van Veen for his contribution providing results for RadAdapt summarization models. We also thank the institutions that have provided funding for this work. Research reported in this publication was made possible in part by the National Institute of Biomedical Imaging and Bioengineering (NIBIB) of the National Institutes of Health under contracts 75N92020C00008 and 75N92020C00021, GE Healthcare, and Stanford Knight Hennessy Scholars.

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

# A   MIMIC-III procedure selection dataset

The MIMIC-III dataset consists of electronic health record data of individuals admitted to critical care units at a large tertiary care hospitals [29]. Within the data made in MIMIC-III are clinical notes, among which are 500,000+ radiology reports. Using regular expressions on the report titles, we select 97,304 containing *CT* in the title. From the report header, the procedure title is extracted and procedure titles are grouped into a set of distinct procedure titles. These were manually reviewed by an MD and a board-certified radiologist, and mapped to the LOINC/RSNA Radiology Playbook[2], a standardized naming and coding convention for 1,000 commonly performed radiology procedures. Each report was then labeled with an corresponding LOINC/RSNA procedure code. Procedure codes that appeared less than 100 times in the corpus were aggregated into an "Other category". This led to a total of 45 categories.

To obtain the source data for each report, regular expressions were used to extract the *Clinical History* section of the report. The extracted segment of text is used as a proxy for the *Reason for Exam*, the motivating clinical context expressed by the clinician when ordering a radiology procedure. The task consists of mapping a free-text *Reason for Exam* to a corresponding procedure code.

A stratified sampling strategy based on the report label was used to obtain final train/dev/test splits consisting of 60/20/20 percent of the total reports, respectively.

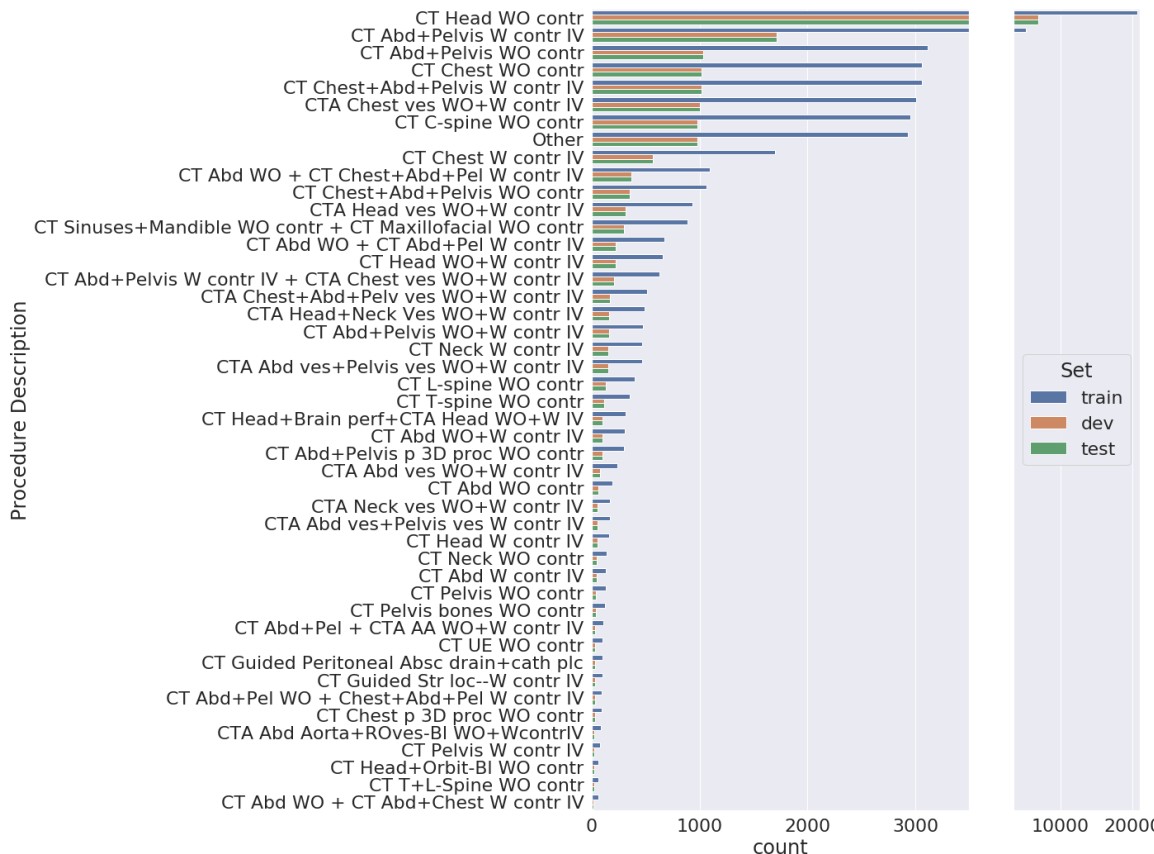

Figure 4: Distribution of CT procedure codes in MIMIC-III procedure selection task.

Figure 4 illustrates the distribution of report labels in the dataset. We note that the majority of entries correspond to a minority of classes, with *CT Head without contrast* being the most common class. Notably, there is a long tail of procedures, motivating the stratified sampling strategy. Examples of entries in the dataset include: *SEIZURES → CT Head WO contr*, *please eval for residual stones →*

---

[2]https://www.rsna.org/practice-tools/data-tools-and-standards/radlex-radiology-lexicon/procedure-names-radlex-playbook

*CT Abd+Pelvis WO contr*, and *Evaluate colon, status of ischemic colitis.* → *CT Abd+Pelvis W contr IV*.

In addition to stratifying by labels, our provided annotated data provides a separate dev and test sets, consisting of patients that do not appear in the train, or train ∪ dev splits, respectively. The number of patients and documents in each set are detailed in Table 5. Performance of models on the set of previously unseen patients is further examined in I, where in general a decrease in performance compared to the overall test set is observed. We hypothesize that in application settings there may be a subset of patients seen during model training, as well as a subset of patients new to the institution. Therefore, it would be relevant to know how a model will perform on a mix of the two, as well as on unseen patients only as a measure of model generalizability. Thus, we provide two dev/test sets so that this difference can be kept in mind in the model design stage.

Table 5: Summary of data splits for MIMIC III Procedure dataset

| Set | n Patients | n Samples |
|---|---|---|
| Train | 19,811 | 58,091 |
| Dev | 11,467 | 19,364 |
| $\text{Dev}_{\text{new only}}$ | 2,170 | 2,603 |
| Test | 11,428 | 19,364 |
| $\text{Test}_{\text{new only}}$ | 1,497 | 1,746 |

$\text{Dev}_{\text{new only}}$ and $\text{Test}_{\text{new only}}$ are subsets of Dev and Test that only contain documents of patients not present in Train, or Train ∪ Dev splits.

## B  BioNLP Summarization Data

Table 6: Number of reports available in the BioNLP2023 report summarization task.

| Modality/Body Part | Number of Reports | | | |
|---|---|---|---|---|
| | Train | Dev | Test | Total |
| CT Abdomen-pelvis | 12791 | 1598 | 798 | 15187 |
| CT Chest | 10228 | 1277 | 639 | 12144 |
| CT Head | 25121 | 3139 | 1569 | 29829 |
| CT Neck | 911 | 113 | 56 | 1080 |
| CT Sinus | 0 | 0 | 633 | 633 |
| CT Spine | 4413 | 550 | 275 | 5238 |
| MR Abdomen | 0 | 0 | 530 | 530 |
| MR Head | 5850 | 730 | 365 | 6945 |
| MR Neck | 0 | 0 | 114 | 114 |
| MR Pelvis | 0 | 0 | 126 | 126 |
| MR Spine | 0 | 0 | 1410 | 1410 |

## C  Additional experimental details

For each DyGIE++ model fine-tuning and evaluation, we leverage the implementation by the original authors[3], which is based on the allennlp and Pytorch libraries. We perform a grid hyperparameter search varying the learning rate (1e-2 - 1e-4), weight decay (0-1), and fine-tuning vs. freezing the language model layers. We use a fixed seed, 2 hidden graph layers of dimension 150, and keep other hyperparameters as described in [51].

For classification and Stanza NER, we leverage the HuggingFace transformers library[4]. We perform hyperparameter optimization using a Tree-structured Parzen Estimator algorithm implemented by Optuna [1]. Using a fixed seed, we perform up to 10 trials varying the learning rate (1e-5 - 1e-4), number of epochs (3-5), batch size (32, 64, 128) and weight decay (1e-12 - 1e-1). We use the

---

[3]https://github.com/dwadden/dygiepp
[4]https://github.com/huggingface/transformers

HuggingFace evaluate library to evaluate models for F1 scores and accuracy. We use the same framework to implement the complementary metrics described in D.

For report summarization, model fine-tuning and evaluation is carried out using the ViLMedic library [13]. Models are trained using an initial learning rate of 5e-5 using the Adam optimizer [33]. A batch size of 32 with gradient accumulations over 0, 2 and 4 batches are explored. The ROUGEL metric is monitored for early stopping (10 epochs). The learning rate is decayed by 0.8 if there is no improvement on the validation set for two epochs.

## D    Complementary metrics for NLU evaluation

It is important to evaluate models on their ability to express their uncertainty and calibration that make them suitable for deployment. Calibration ensures that estimated class probabilities match their naturally occurring prevalence. It is typically measured using the scalar summary statistic, **Expected Calibration Error** (ECE) [41]. Sorted predictions are divided into multiple bins (usually 10) and the absolute difference between the average accuracy and average confidence in the bin is termed the calibration gap. ECE is the weighted average of the calibration gaps [21]. **Static Calibration Error** (SCE) [42] was proposed as an extension to ECE to include every class in the multi-class setting, since ECE uses only the maximum probability and ignores the other class probabilities. Mathematically, the two quantities are defined as

$$\text{ECE} = \sum_{b=1}^{B} \frac{n_b}{N} \; | \, \text{acc}(b) - \text{conf}(b) \, |$$

$$\text{SCE} = \frac{1}{K} \sum_{k=1}^{K} \sum_{b=1}^{B} \frac{n_{bk}}{N} \; | \, \text{acc}(b, k) - \text{conf}(b, k) \, |$$

where $N$ is the total number of samples, $K$ is the total number of classes, $n_b$ and $n_{bk}$ are number of samples in bin $b$ and for class $k$, respectively.

**Weighted Model Confidence**. We propose Weighted Model Confidence (WMC) as a metric that measures the quality of the model's confidence, as weighted by accuracy. This measure assigns positive weight when it's accurate (rewards confident & accurate predictions) and negative weight when inaccurate (penalizes confident & inaccurate predictions). We define it mathematically to be

$$\text{WMC} = \frac{1}{N} \sum_{i=1}^{N} \text{conf}(i) \, . \, (-1)^{\text{acc}(i)+1}$$

where $\text{conf}(i)$ and $\text{acc}(i)$ denote confidence and accuracy of prediction $i$. The higher the value of this metric, the better the model is at correlating confidence with accuracy.

**Average KL divergence with the Uniform Distribution** (aKLU) measures the how peaked the the predicted distribution is by comparing against a flat, uniform distribution via the KL divergence, averaged over the dataset. Mathematically, it is given by

$$\text{aKLU} = \frac{1}{N} \sum_{i=1}^{N} \text{KL}( \, \hat{p}(y_i | \mathbf{x}_i) \, || \, U \, )$$

where $U$ is the uniform distribution over the classes. Higher values of the metric indicate better discriminative power of the model.

**Average Predictive Entropy** (aPE) measures the entropy of the predicted distribution averaged over the dataset. Lower values of aPE imply lower uncertainty in model predictions. Mathematically, it is defined as

$$\text{aPE} = \frac{1}{N} \sum_{i=1}^{N} \sum_{k=1}^{K} \hat{p}(y_i = k | \mathbf{x}_i) \log \hat{p}(y_i = k | \mathbf{x}_i)$$

### D.1    Results

We evaluate all document classification models on the MIMIC-III Procedure Selection dataset using all the complementary metrics and summarize their performance in Table 7. To provide an overview

of performance across these metrics, we rank the models based on each metric, with 1 being the highest (best) rank and 16 being the lowest. We average each model's rank across the different metrics to get the final Average Rank. We do not directly average these additional scores since, unlike F1 and accuracy scores (Table 3), these are not all percentage-based metrics. Overall, similar to what is observed with other evaluation methods, no single model outperforms all others across all tasks. General domain models have inferior performance than biomedical models, of which BioLinkBERT ranks higher across all metrics. These findings further suggest that both pretraining corpora and vocabulary, as well as improved representation learning objectives during pretraining may lead to improved performance across calibration, uncertainty and prediction quality. Notably, GatorTron, which achieves the highest accuracy, ranks 7.4/16 on average across these metrics. Alternatively, ELECTRA$_{small}$ achieves the best calibration, uncertainty and discriminative scores, yet has the lowest accuracy. Our proposed WMC metric, aimed at rewarding both confidence and accuracy clarifies the discordance: GatorTron achieves the best performance and ELECTRA$_{small}$ the worst. We believe that combining this set of metrics with Accuracy and F1 scores to the benchmark has complementary utility and is an important step towards making RaLEs model comparisons more comprehensive. We hope that these evaluations can guide future model developers to train models that are not just accurate but also well-calibrated with high prediction quality and able to express their uncertainty to enable deployment and their integration into larger systems.

Table 7: Complementary NLU Evaluations for document classification models on the MIMIC-III Procedure selection dataset using multiple families of metrics.

| Model | ECE ↓ | SCE ↓ | aPE ↓ | WMC ↑ | aKLU ↑ | Avg Rank ↓ |
|---|---|---|---|---|---|---|
| BERT$_{base}$ | 0.065 | 0.004 | 0.994 | 0.339 | 3.897 | 10.8 |
| BERT$_{large}$ | 0.045 | 0.003 | 1.112 | 0.332 | 3.385 | 8.2 |
| ROBERTA$_{base}$ | 0.040 | **0.002** | 1.144 | 0.320 | 3.472 | 9 |
| ROBERTA$_{large}$ | 0.048 | 0.003 | 1.087 | 0.329 | 3.368 | 8 |
| ELECTRA$_{small}$ | **0.025** | **0.002** | 1.323 | 0.292 | **3.294** | 7.2 |
| ELECTRA$_{base}$ | 0.069 | 0.004 | 0.994 | 0.324 | 4.558 | 12.6 |
| ELECTRA$_{large}$ | 0.046 | 0.003 | 1.121 | 0.321 | 3.722 | 12 |
| DeBERTa$_{base}$ | 0.046 | 0.003 | 1.077 | 0.340 | 3.438 | 6.4 |
| DeBERTa$_{large}$ | 0.061 | 0.003 | 1.065 | 0.325 | 3.640 | 11 |
| PubMedBERT | 0.050 | 0.003 | 1.035 | 0.343 | 3.669 | 8 |
| BioLinkBERT$_{base}$ | 0.046 | 0.003 | 1.077 | 0.340 | 3.438 | 6.2 |
| BioLinkBERT$_{large}$ | 0.035 | **0.002** | 1.107 | 0.337 | 3.384 | **5.4** |
| BioClinicalBERT | 0.047 | 0.003 | 1.089 | 0.337 | 3.377 | 7 |
| GatorTron | 0.061 | 0.003 | **0.977** | **0.355** | 3.556 | 7.4 |
| RadBERT$_1$ | 0.070 | 0.004 | 0.988 | 0.340 | 3.697 | 10.4 |
| RadBERT$_2$ | 0.052 | 0.003 | 1.059 | 0.343 | 3.415 | 6.4 |

Abbreviations: ECE: Expected Calibration Error, SCE: Static Calibration Error, aPE: average Predictive Entropy, WMC: Weighted model confidence, aKLU: average KL divergence with uniform distribution.

Table 8: Pearson Correlation Coefficient between added complementary metrics on NLU evaluation.

| | ECE | SCE | aPE | WMC | aKLU |
|---|---|---|---|---|---|
| **ECE** | 1.00 | 0.93 | -0.89 | 0.51 | 0.70 |
| **SCE** | 0.93 | 1.00 | -0.72 | 0.25 | 0.78 |
| **aPE** | -0.89 | -0.72 | 1.00 | -0.82 | -0.54 |
| **WMC** | 0.51 | 0.25 | -0.82 | 1.00 | 0.02 |
| **aKLU** | 0.70 | 0.78 | -0.54 | 0.02 | 1.00 |

Abbreviations: ECE: Expected Calibration Error, SCE: Static Calibration Error, aPE: average Predictive Entropy, WMC: Weighted model confidence, aKLU: average KL divergence with uniform distribution.

# E   Data efficiency experiment results

Table 9: Summary of results for NLU tasks using 10% training and validation labels.

| Model | $RG_{NER}$† | $RG_{NER}$‡ | $RG_{RE}$† | $RG_{RE}$‡ | RadSpRL | Stanza | Procedure | Avg |
|---|---|---|---|---|---|---|---|---|
| BERT$_{base}$ | 90.4 | 82.8 | 77.6 | 63.1 | 75.1 | 6.6 | 53.9 | 64.2 |
| BERT$_{large}$ | 89.7 | 79.7 | 76.2 | 62.6 | 60.1 | 7.0 | 58.8 | 62.0 |
| RoBERTa$_{base}$ | 88.8 | 82.2 | 75.1 | 61.5 | 73.2 | 3.2 | 57.3 | 63.0 |
| RoBERTa$_{large}$ | 88.5 | 78.7 | 73.2 | 60.6 | 74.4 | 4.4 | 58.9 | 62.7 |
| ELECTRA$_{small}$ | 88.6 | 81.5 | 73.9 | 57.8 | 68.0 | 5.3 | 48.9 | 60.6 |
| ELECTRA$_{base}$ | 92.1 | 79.7 | 75.5 | 58.2 | 71.5 | 8.2 | 54.7 | 62.8 |
| ELECTRA$_{large}$ | 92.5 | 80.5 | 81.2 | 66.3 | 69.5 | 6.5 | 41.6 | 62.6 |
| DeBERTa-V3$_{base}$ | 92.9 | 88.9 | 75.9 | 62.1 | 78.7 | 4.5 | 54.0 | 65.3 |
| DeBERTa-V3$_{large}$ | 90.7 | 81.6 | 75.5 | 63.0 | 79.1 | 7.0 | 49.1 | 63.7 |
| PubMedBERT | 91.6 | 82.6 | 78.6 | 64.7 | 77.9 | 5.3 | 60.5 | 65.9 |
| BioLinkBERT$_{base}$ | 91.5 | 86.5 | 79.9 | 65.1 | 80.8 | 4.2 | 60.1 | 66.9 |
| BioLinkBERT$_{large}$ | 90.8 | 87.1 | 80.7 | 62.0 | 75.7 | 6.0 | 57.8 | 65.7 |
| BioClinicalBERT | 91.5 | 86.7 | 75.9 | 59.4 | 80.0 | 4.4 | 60.3 | 65.5 |
| GatorTron | 92.2 | 87.7 | 80.6 | 66.7 | 81.8 | 4.3 | 62.6 | 68.0 |
| RadBERT1 | 91.1 | 84.5 | 73.1 | 62.8 | 71.3 | 7.2 | 60.9 | 64.4 |
| RadBERT2 | 91.1 | 85.3 | 74.0 | 62.6 | 71.2 | 3.3 | 59.6 | 63.9 |

†: MIMIC-CXR (in domain), ‡: CheXpert (out of domain).

Table 10: Summary of results for NLU tasks using 1% training and validation labels.

| Model | RadGraph | | | | RadSpRL | Stanza | Procedure | Avg |
|---|---|---|---|---|---|---|---|---|
| | $RG_{NER}$† | $RG_{NER}$‡ | $RG_{RE}$† | $RG_{RE}$‡ | | | | |
| BERT$_{base}$ | 0.0 | 0.0 | 0.0 | 0.0 | 0.0 | 6.5 | 48.1 | 9.1 |
| BERT$_{large}$ | 2.2 | 1.9 | 0.0 | 0.0 | 0.0 | 4.0 | 40.3 | 7.7 |
| RoBERTa$_{base}$ | 0.0 | 0.0 | 0.0 | 0.0 | 0.0 | 1.0 | 39.6 | 6.8 |
| RoBERTa$_{large}$ | 0.0 | 0.0 | 0.0 | 0.0 | 0.0 | 4.4 | 45.5 | 8.3 |
| ELECTRA$_{small}$ | 1.5 | 1.1 | 0.0 | 0.0 | 0.0 | 6.0 | 35.6 | 7.1 |
| ELECTRA$_{base}$ | 0.0 | 0.0 | 0.0 | 0.0 | 0.0 | 8.7 | 38.8 | 7.9 |
| ELECTRA$_{large}$ | 0.0 | 0.0 | 0.0 | 0.0 | 0.0 | 7.0 | 35.6 | 7.1 |
| DeBERTa-V3$_{base}$ | 0.0 | 0.0 | 0.0 | 0.0 | 0.9 | 6.1 | 39.1 | 7.7 |
| DeBERTa-V3$_{large}$ | 0.0 | 0.0 | 0.0 | 0.0 | 0.0 | 7.2 | 38.9 | 7.7 |
| PubMedBERT | 0.0 | 0.0 | 0.0 | 0.0 | 0.0 | 1.1 | 40.4 | 6.9 |
| BioLinkBERT$_{base}$ | 0.0 | 0.0 | 0.0 | 0.0 | 0.0 | 4.1 | 35.6 | 6.6 |
| BioLinkBERT$_{large}$ | 0.0 | 0.0 | 0.0 | 0.0 | 0.0 | 6.3 | 37.2 | 7.3 |
| BioClinicalBERT | 0.0 | 0.0 | 0.0 | 0.0 | 0.0 | 5.8 | 45.2 | 8.5 |
| GatorTron | 0.4 | 0.4 | 0.0 | 0.0 | 0.0 | 5.2 | 55.4 | 10.2 |
| RadBERT1 | 0.1 | 0.1 | 0.0 | 0.0 | 0.0 | 7.4 | 37.8 | 7.7 |
| RadBERT2 | 2.4 | 2.0 | 0.0 | 0.0 | 0.0 | 1.3 | 52.0 | 9.2 |

†: MIMIC-CXR (in domain), ‡: CheXpert (out of domain).

# F  Linear probe experiment results

Table 11: Summary of results for NLU tasks keeping language model weights frozen during fine-tuning.

| Model | RG$_{NER}$† | RG$_{NER}$‡ | RG$_{RE}$† | RG$_{RE}$‡ | RadSpRL | Stanza | Procedure | Avg |
|---|---|---|---|---|---|---|---|---|
| BERT$_{base}$ | 91.4 | 88.9 | 77.1 | 65.7 | 87.4 | 39.4 | 35.7 | 69.4 |
| BERT$_{large}$ | 91.9 | 88.5 | 72.4 | 57.7 | 82.5 | 13.3 | 35.6 | 63.1 |
| RoBERTa$_{base}$ | 89.3 | 85.8 | 70.5 | 56.3 | 82.9 | 12.5 | 35.6 | 61.8 |
| RoBERTa$_{large}$ | 89.1 | 83.9 | 56.4 | 39.9 | 83.4 | 18.3 | 37.5 | 58.3 |
| ELECTRA$_{small}$ | 88.1 | 80.3 | 69.7 | 52.6 | 86.3 | 12.5 | 35.8 | 60.8 |
| ELECTRA$_{base}$ | 91.1 | 87.1 | 77.0 | 60.1 | 90.0 | 37.6 | 39.0 | 68.8 |
| ELECTRA$_{large}$ | 90.8 | 85.7 | 69.5 | 56.5 | 88.1 | 33.5 | 38.1 | 66.0 |
| DeBERTa-V3$_{base}$ | 90.2 | 86.6 | 49.4 | 32.0 | 85.2 | 22.7 | 35.6 | 57.4 |
| DeBERTa-V3$_{large}$ | 89.9 | 84.8 | 55.4 | 38.8 | 87.3 | 23.0 | 35.6 | 59.3 |
| PubMedBERT | 92.9 | 89.4 | 82.1 | 70.5 | 88.5 | 21.9 | 41.8 | 69.6 |
| BioLinkBERT$_{base}$ | 92.1 | 90.5 | 79.0 | 68.0 | 90.5 | 10.4 | 35.6 | 66.6 |
| BioLinkBERT$_{large}$ | 92.2 | 90.7 | 74.9 | 63.5 | 85.4 | 43.5 | 38.3 | 69.8 |
| BioClinicalBERT | 92.3 | 89.5 | 78.9 | 66.0 | 88.3 | 34.9 | 35.6 | 69.4 |
| GatorTron | 93.6 | 89.5 | 81.1 | 68.1 | 91.2 | 42.0 | 49.3 | 73.5 |
| RadBERT1 | 91.7 | 87.8 | 76.2 | 61.6 | 88.0 | 16.5 | 51.0 | 67.5 |
| RadBERT2 | 91.3 | 88.2 | 74.3 | 60.8 | 89.1 | 19.7 | 50.4 | 67.7 |

†: MIMIC-CXR (in domain), ‡: CheXpert (out of domain).

# G   Out of distribution NLG evaluation

Table 12: NLG evaluation on out of distribution (Stanford) and in distribution (Indiana) test sets.

| Model | Stanford | | Indiana | |
|---|---|---|---|---|
| | R-2 | RG | R-2 | RG |
| ELECTRA$_{base}$ | .232 | .382 | .244 | .374 |
| BioLinkBERT$_{base}$ | .227 | .397 | **.263** | .385 |
| GatorTron | **.238** | **.408** | .262 | **.404** |
| RadBERT$_2$ | .222 | .383 | .251 | .379 |
| Prior SOTA† | **.277** | - | **.596** | - |
| Prior Baseline‡ | .241 | - | .287 | - |

†:[10], ‡:from [3]

# H   Report Length Analysis

Table 13: Report length statistics for the report summarization datasets. Note that only findings and impressions sections were used to calculate report lengths.

| Dataset | Split | Min | Max | Average | Standard Deviation |
|---|---|---|---|---|---|
| BioNLP2023 | train | 8 | 1279 | 157.5 | 85.3 |
| | dev | 15 | 964 | 158.6 | 87.1 |
| | test | 16 | 992 | 166.2 | 89.2 |
| MEDIQA2021 | train | 19 | 430 | 63.5 | 31.0 |
| | dev | 21 | 240 | 63.2 | 31.0 |
| | dev$_{indiana}$ | 15 | 230 | 43.1 | 19.7 |
| | test | 18 | 220 | 57.2 | 31.6 |

# I   Procedure Selection Error Analysis

The following tables and figure outline error analysis for the best performing model in the benchmark, a fine-tuned GatorTron model. Evaluation metrics are obtained by studying this model across the MIMIC III Procedure selection test set.

Table 14 shows that this model may be confidently correct and incorrect. This was also quantified in Table 7, which shows metrics such as ECE. Further, example indications such as "Please evaluate for interval progression of hemorrhage" reveal that there are instances where the provided indication alone may not be sufficient to reliable determine the ground truth protocol. This observation suggests the opportunity to develop improved datasets in the future, possibly integrating prior clinical notes with the examination indications.

Table 15 shows that model performance varies by class label availability, with underrepresented classes having poorer performance in general. However, as visualized in Figure 5, this is not universally the case, as some underrepresented classes exhibit relatively higher performance with lower label availability. We hypothesize that certain classes with highly specific indications are easier to classify, even with smaller sample sizes in the training data.

Finally, Figure 6 shows the drop in performance when evaluating only new patients in the test set. The drop is observed across all model types, label availabilities and tuning strategies (full fine-tuning vs. linear probe).

Table 14: Examples of correct and incorrect predictions made by the best-performing classifier in the procedure prediction task.

| Indication | Reference | Predicted | Correct? | Ref $P$ | Pred $P$ |
|---|---|---|---|---|---|
| evaluate s/p drainage of SDH | CT Head WO contr | CT Head WO contr | ✓ | 0.99 | 0.99 |
| evaluate for progression of subdural hematomas | CT Head WO contr | CT Head WO contr | ✓ | 0.99 | 0.99 |
| eval for hemorrhage post-op, 4 hours after surgery | CT Head WO contr | CT Head WO contr | ✓ | 0.99 | 0.99 |
| R/o IPMN | CT Abd WO+W contr IV | CT Head WO contr | ✗ | 0.00 | 0.99 |
| Please evaluate for interval progression of hemorrhage. | CT Abd+Pelvis p 3D proc WO contr | CT Head WO contr | ✗ | 0.00 | 0.99 |
| r/o met, bleed | CT T+L-Spine WO contr | CT Head WO contr | ✗ | 0.00 | 0.94 |

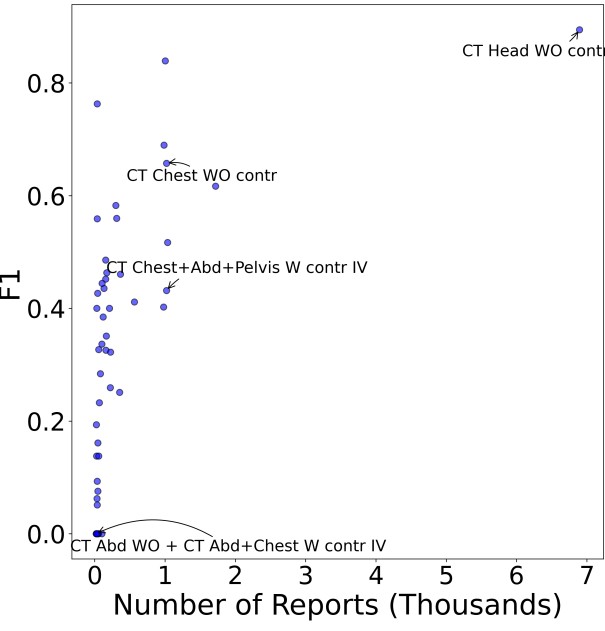

Figure 5: Per-class performance as a function of number of reports.

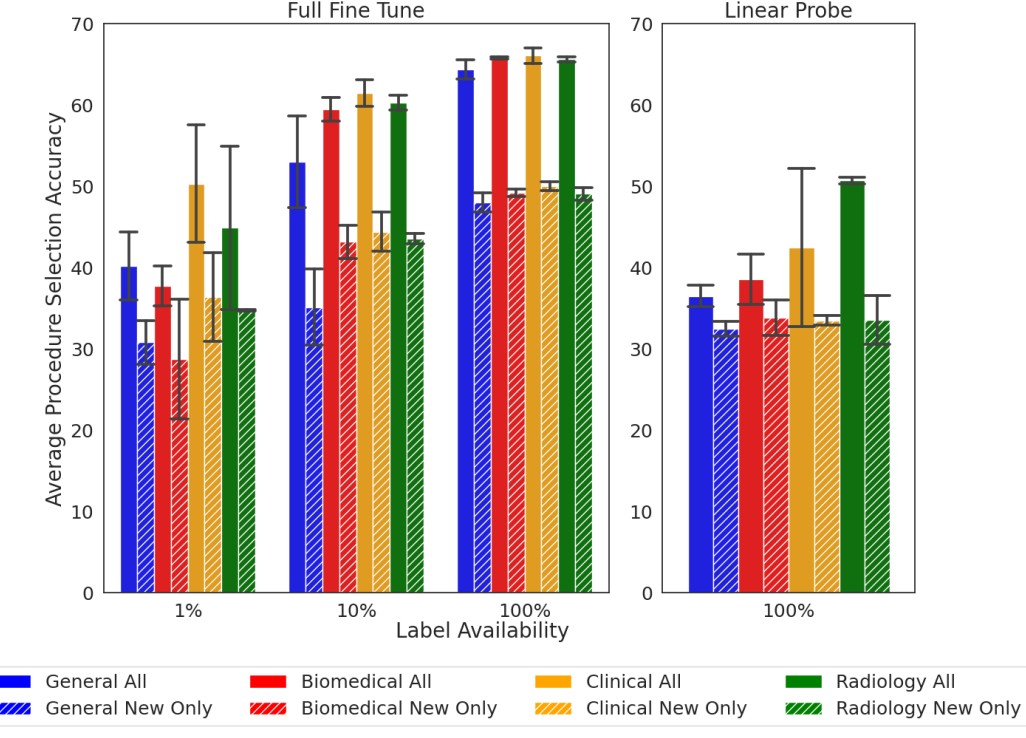

Figure 6: Effect of excluding patients seen during training from test set. Error bars indicate standard deviation across the models within each category.

Table 15: Per-class performance for procedure prediction task for model with highest overall accuracy. Number of samples are reported in the test set.

| Procedure | Precision | Recall | F1 | N |
|---|---|---|---|---|
| CT Head WO contr | 84.3 | 95.3 | 89.4 | 6894 |
| CTA Chest ves WO+W contr IV | 80.8 | 87.3 | 83.9 | 1002 |
| CT Guided Str loc–W contr IV | 69.0 | 85.3 | 76.3 | 34 |
| CT C-spine WO contr | 68.5 | 69.5 | 69.0 | 984 |
| CT Chest WO contr | 64.3 | 67.3 | 65.7 | 1021 |
| CT Abd+Pelvis W contr IV | 54.5 | 71.1 | 61.7 | 1717 |
| CT Sinuses+Mandible WO contr + CT Maxillofacial WO contr | 67.4 | 51.3 | 58.3 | 298 |
| CTA Head ves WO+W contr IV | 57.9 | 54.2 | 56.0 | 312 |
| CT UE WO contr | 55.9 | 55.9 | 55.9 | 34 |
| CT Abd+Pelvis WO contr | 56.7 | 47.5 | 51.7 | 1037 |
| CT Neck W contr IV | 48.4 | 48.7 | 48.6 | 156 |
| CTA Chest+Abd+Pelv ves WO+W contr IV | 43.1 | 50.0 | 46.3 | 170 |
| CT Abd WO + CT Chest+Abd+Pel W contr IV | 41.5 | 51.6 | 46.0 | 366 |
| CTA Abd ves+Pelvis ves WO+W contr IV | 38.5 | 54.8 | 45.2 | 155 |
| CT Head+Brain perf+CTA Head WO+W IV | 69.4 | 32.7 | 44.4 | 104 |
| CT L-spine WO contr | 52.7 | 37.1 | 43.6 | 132 |
| CT Chest+Abd+Pelvis W contr IV | 42.1 | 44.3 | 43.2 | 1020 |
| CT Pelvis bones WO contr | 45.7 | 40.0 | 42.7 | 40 |
| CT Chest W contr IV | 45.8 | 37.3 | 41.1 | 565 |
| Other | 49.8 | 33.7 | 40.2 | 978 |
| CT Abd+Pelvis W contr IV + CTA Chest ves WO+W contr IV | 55.0 | 31.4 | 40.0 | 210 |
| CTA Abd Aorta+ROves-Bl WO+WcontrIV | 52.9 | 32.1 | 40.0 | 28 |
| CT T-spine WO contr | 44.4 | 33.9 | 38.5 | 118 |
| CTA Head+Neck Ves WO+W contr IV | 45.6 | 28.5 | 35.1 | 165 |
| CT Abd WO+W contr IV | 32.1 | 35.3 | 33.6 | 102 |
| CTA Neck ves WO+W contr IV | 36.2 | 29.8 | 32.7 | 57 |
| CT Abd+Pelvis WO+W contr IV | 41.0 | 27.0 | 32.6 | 159 |
| CT Abd WO + CT Abd+Pel W contr IV | 31.7 | 32.7 | 32.2 | 223 |
| CTA Abd ves WO+W contr IV | 29.3 | 27.5 | 28.4 | 80 |
| CT Head WO+W contr IV | 31.0 | 22.3 | 25.9 | 220 |
| CT Chest+Abd+Pelvis WO contr | 44.6 | 17.5 | 25.1 | 355 |
| CT Abd WO contr | 45.5 | 15.6 | 23.3 | 64 |
| CT Head+Orbit-Bl WO contr | 33.3 | 13.6 | 19.4 | 22 |
| CT Pelvis WO contr | 26.3 | 11.6 | 16.1 | 43 |
| CT Pelvis W contr IV | 1 | 7.4 | 13.8 | 27 |
| CTA Abd ves+Pelvis ves W contr IV | 18.8 | 10.9 | 13.8 | 55 |
| CT Abd+Pel + CTA AA WO+W contr IV | 25.0 | 5.7 | 9.3 | 35 |
| CT Neck WO contr | 25.0 | 4.4 | 7.5 | 45 |
| CT Chest p 3D proc WO contr | 1 | 3.2 | 6.2 | 31 |
| CT Guided Peritoneal Absc drain+cath plc | 20.0 | 2.9 | 5.1 | 34 |
| CT T+L-Spine WO contr | 0 | 0 | 0 | 22 |
| CT Abd W contr IV | 0 | 0 | 0 | 45 |
| CT Abd WO + CT Abd+Chest W contr IV | 0 | 0 | 0 | 19 |
| CT Abd+Pel WO + Chest+Abd+Pel W contr IV | 0 | 0 | 0 | 32 |
| CT Abd+Pelvis p 3D proc WO contr | 0 | 0 | 0 | 101 |
| CT Head W contr IV | 0 | 0 | 0 | 53 |

## J   Report Summarization Errors

Table 16 lists example reference and generated impressions for MEDIQA2021 and BioNLP2023, the report summarization tasks examined in RaLEs. We highlight the limitations of current automated evaluation approaches. For example, when additional recommendations based on medical expertise are provided in the hypothesis, metrics such as R-L impose additional penalty on the prediction if these are not present in the reference. Conversely, in scenarios where no specific disease entities are mentioned in the ground truth, metrics like RG unfairly penalize the models given that they rely on the explicit mention of diseases and their relation to anatomies. The RG metric may be further affected by the lack of validation of RadGraph labeler on non-Chest-X-ray reports, as well as lack of normalization to a standardized nomenclature for entities. Such examples illustrate the opportunity for improved summarization evaluations. Further, they motivate importance of developing models that not only understand the immediate report but can also draw upon broader context, enabling them to offer recommendations rooted in both the findings and best clinical practice.

Table 16: Report summarization errors.

| Dataset | Findings | Reference Impression | Generated Impression | R-L | RG | CheXbert | NLG |
|---|---|---|---|---|---|---|---|
| | __ and lateral views of the chest demonstrate no focal infiltrate, pleural effusion, or pneumothorax. No bony abnormality is seen. The right scapula appears intact. No acute fractures are seen. prediction | No acute cardiopulmonary process. No acute fracture is seen | no acute cardiopulmonary process no acute fracture is identified if there is continued concern for rib fracture a dedicated rib series is recommended | 0.50 | 0.89 | 0.0 | 0.695 |
| MEDIQA 2021 | Status post posterior spinal fusion. Status post ____ sternotomy. Stable position of the cardiac ____ and leads. The cardiac silhouette and mediastinum size are normal. The lungs are clear. prediction | Stable position of the cardiac ____ and leads. Normal cardiac size with clear lungs. | no acute cardiopulmonary process | 0.0 | 0.0 | 0.67 | 0.0 |
| | Heart and mediastinal contours are unremarkable. The pulmonary vasculature is normal in appearance. The lung parenchyma is clear, without focal infiltrate. There are no pleural effusions, and there is no pneumothorax. The visualized bony structures are grossly unremarkable. No displaced rib fractures. Right nipple ring noted. | no acute radiographic cardiopulmonary process. | no acute pulmonary process | 0.67 | 0 | 1 | 0.335 |
| | the patient is status post surgical intervention on the left side as evidenced by left frontoparietal craniotomy with drainage catheters. there is interval decrease in the size of the left convexity subdural hematoma; however, there remains high hyperintensity material along the left cerebral convexity. the appearance of the interhemispheric fissure subdural hematoma appears stable. there is less rightward shift of normally midline structures compared to prior examination and there is associated reexpansion of the lateral ventricles. there is new finding of left and right frontal pneumocephalus which is related to surgical intervention. there is also subcutaneous gas adjacent to the left masseter muscle and within the left frontal and temporal subcutaneous tissue. there is also significant soft tissue swelling adjacent to the left craniotomy. there are no new areas of hemorrhage. there are no areas of infarction. the sinuses are clear. | no evidence of intracranial hemorrhage or mass effect. | no evidence of intracranial hemorrhage or mass effect | 1.00 | 1.00 | - | 1 |
| BioNLP 2023 | the right basal ganglionic parenchymal hemorrhage and surrounding vasogenic edema is unchanged since the ___ study, again measuring 1.9 x 3.5 cm. small intraventricular component layering in the bilateral occipital horns has resolved. there is no new focus of hemorrhage, large territorial infarction, mass, or shift of normally midline structures. asymmetry of all the components of the right lateral ventricle, unchanged, is likely congenital/developmental. increased opacification of the bilateral sphenoid air cells with airfluid levels, which may relate to supine positioning. the mastoid air cells are well pneumatized. | 1) no evidence of acute hemorrhage. 2) periventricular white matter changes are consistent with chronic microvascular infarctions. 3) involutional change of the brain. 4) hypodense lesion adjacent to the left internal capsule likely represents an old infarct. | no evidence of intracranial hemorrhage or mass effect | 0.18 | 0.0 | - | 0.09 |
| | there is no intracranial mass lesion, hydrocephalus, shift of normally midline structures, major vascular territorial infarct, or intracranial hemorrhage. other than bifrontal small, ill- defined low-attenuation foci (eg. 2:17-19), which may represent chronic microvascular infarction, the ___- white matter differentiation is preserved. the osseous and soft tissue structures are unremarkable. the visualized paranasal sinuses are clear. | 1. no evidence of acute hemorrhage or mass effect. 2. marked calcification of the carotid and vertebral arteries, more on the right than the left. | no evidence of intracranial hemorrhage or mass effect | 0.41 | 0.0 | - | 0.20 |

