# OpenReview forum: "RaLEs: a Benchmark for Radiology Language Evaluations"
_NeurIPS.cc/2023/Track/Datasets_and_Benchmarks — NeurIPS 2023 Datasets and Benchmarks Poster_

### Official Review · Reviewer_mpiG · 2023-06-23
**Review of RaLEs for radiology benchmarking**

**Rating:** 5
**Confidence:** 4
**Correctness:** Nothing of note.
**Clarity:** There are no major concerns with clar…

**Strengths:**

The strength of this work is the compilation of multiple publicly available data sets (plus the addition of one small newly de-identified data set) to evaluate multiple different tasks required to utilize NLP methods for evaluation of radiology reports, including named entity recognition, relationship extraction, and summarization.  Often these tasks are separated in works.  Thus, including a presentation with all tasks is a key strength.

The work chooses a comprehensive set of state-of-the-art models from to perform the evaluation and create the benchmark.  The selection of methods including multiple specialty BERT models for biomedical, clinical, and radiology corpuses; various Electra models; and Roberta models.  As such, the authors did select the most current models likely to succeed for the presented tasks.

The authors create a code framework for data standardization to help streamline future development.



**Additional Feedback:**

The paper is presently more like a review using existing data sets with existing models.  It does not really fit the traditional research expected in the NeurIPS dataset and benchmark track. From a benchmark standpoint, there does not appear to be a stand-alone code for a benchmark to exactly reproduce the results given limitations on access to the data utilized for the project.  Additionally, from a data set standpoint, the amount of new data generated is practically negligible, and even that data will still have major restrictions.

**Documentation:**

New data not available at this time.  Code for the benchmark will be made public but will not have integrated data.  A bit more documentation and examples are necessary to insure the same data [which cannot be integrated] can be processed and reproduced at a later date to use this work as a possible benchmark moving forward.

**Ethics:**

Nothing of note.

**Limitations:**

The authors address most limitations.  The ability to create a true benchmark without integrated data is a major limitation. While the authors may not be able to fix this directly, it should be acknowledged in an effort to improve access moving forward.

**Opportunities For Improvement:**

Particularly for a data set track, the authors are not providing much in terms of new data sets.  There is a small data set (about 150 patients) that was de-identified and is intended to made a available with some research protocols.

The data curation for the newly de-identified data set does not provide sufficient details on curation and quality control processes.

The authors create code with a suite of models.  However, because there is no direct access to data sets, it is not clear how useful the framework will be for future researchers as a true benchmark.

The figures are of low quality, especially Figure 2.  The scale makes it difficult to view.  Getting rid of grid lines, cleaning up the grey outer boxes, and increasing resolution would help.  Also, the full fine-tune model should be separated from the linear probe using panels in order to minimize confusion. While error bars are shown in Figure 2, there is no quantitative analysis to illustrate whether there are statistically significant differences in performance.

Many radiology reports have unbalanced class data.  There is little to no mention of how authors dealt with class imbalance during pre-processing and in model evaluation.

The authors present data based on anatomical location (Figure 4).  It would be interesting to compare the chest only results (where there is much more data) compared to results for the more sparse anatomical locations.

There is no statistics for the generalizability of frameworks across different data sets.



**Relation To Prior Work:**

The authors should review the recently archived Radiology-GPT as it relevant to their current work.  Also, some of the statements about a lack of benchmark for radiology are exaggerated (line 79).  The authors should spend a bit more comparing their work to that of Casey, et. al. 2021 which did a review of NLP models for radiology.

**Summary And Contributions:**

The authors create a framework for benchmarking radiology language evaluations.  RaLEs is comprised of the following data sets: RadGraph, RadSpRL, Stanza Radiology, MIMIC-III, MEDIAQA 2021 report summarization and BioNLP 2023 report summarization.  A host of standard pre-trained models were utilized for named entity recognition, relationship extraction. and abstractive summarization.  Accuracy results are reported to compare and contrast methods as part of the creation of a benchmark for future development.  The work also presents a natural language understanding (NLU) and natural language generation (NLG) benchmark for the radiology domain.

---

> ### Author Response · Authors · 2023-08-21
> **Thank you for your review and suggestions for improvement (part 1/2).**
>
> > Particularly for a data set track, the authors are not providing much in terms of new data sets. There is a small data set (about 150 patients) that was de-identified and is intended to made a available with some research protocols.
>
> Based on your feedback, we are in the process of annotating 500 new RadGraph reports from our institution. We are finalizing our quality checks and will update the manuscript with the updated dataset and results once they are available.
> We also note that the MIMIC III procedure selection dataset is newly introduced in this work. We have made this more clear in updated Table 1. Further, our work aims to go beyond providing new data and in addition enables benchmarking of language models across a variety of radiology settings.
>
> >The data curation for the newly de-identified data set does not provide sufficient details on curation and quality control processes.
>
> We provide additional details as requested. Our data was acquired at a large academic hospital, and extracted under IRB review. Labels for the dataset were obtained through board-certified radiologists. Full details of the annotation process are described in Zhang et al [1]. For the de-identification, we used state-of-the-art hidden in plain sight de-identification [2], followed by human inspection of each individual report to ensure no original PHI was missed by the algorithm.
>
> [1]Zhang, Y., Zhang, Y., Qi, P., Manning, C. D., & Langlotz, C. P. (2021). Biomedical and clinical English model packages for the Stanza Python NLP library. Journal of the American Medical Informatics Association, 28(9), 1892-1899.
> [2]Chambon, P. J., Wu, C., Steinkamp, J. M., Adleberg, J., Cook, T. S., & Langlotz, C. P. (2023). Automated deidentification of radiology reports combining transformer and “hide in plain sight” rule-based methods. Journal of the American Medical Informatics Association, 30(2), 318-328.
>
> >The authors create code with a suite of models. However, because there is no direct access to data sets, it is not clear how useful the framework will be for future researchers as a true benchmark. [..] The ability to create a true benchmark without integrated data is a major limitation.
>
> Thank you for your suggestions. We have improved our documentation as to how users may use the data and code to reproduce these results. Further, we will release the data under a direct download link [here](https://vilmedic.app/datasets/papers), allowing users with appropriate credentials to access the data without needing to process it. We will update this comment once the data is available at the link, which we expect will be before the end of the discussion period.
>
> >The figures are of low quality, especially Figure 2. [...] there is no quantitative analysis to illustrate whether there are statistically significant differences in performance.
>
> Thank you for your suggestions for improving the quality of our figures. We have incorporated them into the revised manuscript as advised. We did not find statistically significant differences in performance across model categories within the same label availability, and have noted this in the manuscript.
>
> >Many radiology reports have unbalanced class data. There is little to no mention of how authors dealt with class imbalance during pre-processing and in model evaluation. The authors present data based on anatomical location (Figure 4). It would be interesting to compare the chest only results (where there is much more data) compared to results for the more sparse anatomical locations.
>
> In the current version of the work, we use a cross entropy loss for optimization. Potential alternatives may consider data resampling strategies or loss weighting to favor underrepresented classes. In our newly introduced error analysis for the procedure selection task, which suffers from the biggest class imbalance, we observe that class imbalance is only partially responsible for disparate performance across classes (new Figure 5, Table 14). We consider that future work can evaluate whether the aforementioned alternative strategies result in improved performance on minority classes without affecting overall performance.
>
> >There is no statistics for the generalizability of frameworks across different data sets.
>
> Ideally we’d like to evaluate the generalizability across datasets. Currently, only the RadGraph dataset includes an analysis of generalizability between MIMIC-CXR and CheXpert reports. We also aim to include new analysis of generalizability leveraging our newly annotated RadGraph dataset, which will include non-chest x-ray reports that do not come from MIMIC. As future datasets are released, they can be incorporated into our framework, and we have provided instructions as to how to do that in our updated documentation (see [here](https://github.com/StanfordMIMI/RaLEs/tree/main/datasets#adding-a-dataset-to-the-rales-benchmark)).

---

> > ### Author Response · Authors · 2023-08-28
> > **The data download link is now live.**
> >
> > We have now made the data download link live. It is available here https://vilmedic.app/datasets/papers.

---

> ### Author Response · Authors · 2023-08-21
> **Thank you for your review and suggestions (response part 2/2)**
>
> >The authors should review the recently archived Radiology-GPT as it relevant to their current work.
>
> Based on your suggestion we have benchmarked RadiologyGPT in our summarization tasks, given that this is the only task that this model is trained for. The results are in updated Table 4. The current evaluation reveals the shortcoming of zero-shot RadiologyGPT across our two datasets (RaLEs NLG score of 0.18). We are also include results for RadAdapt[1], a state of the art summarization model on BioNLP2023, and are in the process of evaluating this approach for MEDIQA2021. In the future, we plan to continue benchmarking of other zero-shot approaches and parameter efficient adaptation methods.
>
> [1] Dave Van Veen, Cara Van Uden, Maayane Attias, Anuj Pareek, Christian Bluethgen, Malgorzata Polacin, Wah Chiu, Jean-Benoit Delbrouck, Juan Zambrano Chaves, Curtis Langlotz, Akshay Chaudhari, and John Pauly. 2023. RadAdapt: Radiology Report Summarization via Lightweight Domain Adaptation of Large Language Models. In The 22nd Workshop on Biomedical Natural Language Processing and BioNLP Shared Tasks, pages 449–460, Toronto, Canada. Association for Computational Linguistics.
>
> >Also, some of the statements about a lack of benchmark for radiology are exaggerated (line 79). The authors should spend a bit more comparing their work to that of Casey, et. al. 2021 which did a review of NLP models for radiology.
>
> We better acknowledge the valuable contribution of Casey et. al. in our Related Works section. At the same time, we point out that Casey et. al. carry out a systematic review of the literature, without directly evaluating any models. The gap that our work aims to address is to compare the performance of different existing models across a broad range of tasks. We have made this point more clear in line 79.
>
> >The paper is presently more like a review using existing data sets with existing models. It does not really fit the traditional research expected in the NeurIPS dataset and benchmark track. From a benchmark standpoint, there does not appear to be a stand-alone code for a benchmark to exactly reproduce the results given limitations on access to the data utilized for the project. Additionally, from a data set standpoint, the amount of new data generated is practically negligible, and even that data will still have major restrictions.
>
> We respectfully point out that the call for the datasets and benchmarks track includes within its scope “New datasets, or carefully and thoughtfully designed (collections of) datasets based on previously available data”, in addition to “Benchmarks on new or existing datasets, as well as benchmarking tools.”. Our contribution introduces new datasets, accompanied by a thoughtfully designed collection of additional datasets, in addition to providing benchmarking tools and previously unknown results. Hence, we consider our contribution goes beyond that of a review.
> As mentioned above, we believe our new direct (assuming appropriate credentials) dataset access and improved documentation, along with planned release of fine-tuned models, will facilitate the reproduction of our results.

---

> ### Comment · Reviewer_mpiG · 2023-08-29
> **Re-evaluation of RaLEs revisions**
>
> The authors have done work to improve clarity and writing. However, some of the technical concerns still remain.  I stand by my original recommendation of "marginally below acceptance threshold".

---

> > ### Author Response · Authors · 2023-08-29
> >
> > In the rebuttal, we have provided additional technical details and clarifications for the topics covered in the review, including: details on data curation, providing direct data access, updated figures, per-class error analyses, implementation and inclusion of RadiologyGPT evaluations.
> > If the reviewer could be more specific about which concerns still persist, we would be happy to engage in further discussion.

---

### Official Review · Reviewer_SKzi · 2023-07-20
**Curated benchmark of various radiology NLP datasets to ensure LM performance comparisons**

**Rating:** 6
**Confidence:** 3
**Correctness:** N/A

**Strengths:**

- This paper curates radiology text datasets across diverse NLP tasks

- The authors provide configs + fine-tuning scripts to replicate their results across all the models they tested. Have the authors ensured the random seed is fixed as well? I see the `training_args.seed` but I couldn't find that within the config files (presumably this is a fixed internal default though).


**Additional Feedback:**

N/A

**Clarity:**

The paper generally reads well and straightforwardly.

Minor note - the citation style is somewhat hard to parse in the text; this may be due to being a draft, but it would be easier to follow if referenced as brackets to a number in the references. For many cases, however, it's very clearly within the end of a sentence, or an obvious subject of a sentence, hence this note can be ignored.

**Documentation:**

The repo still looks under construction; documentation seems quite sparse. I would appreciate if the authors would provide the following instructions:

1. Clear instructions on how to compile the benchmark dataset (I see there are scripts but I don't recall them linked in the top level README)

2. A tutorial on how to bring a model to test across different datasets in the benchmark

3. Are there plans for a leaderboard to reveal performance across each category, and in general? What are the ways new users can update the leaderboard, and how frequently will this be updated?

4. Plans for maintenance, improvement, or longevity of the package?

The above should be proposed on the main README of the repo (or at least the main README should contain supporting links to access this information).

Also, do the authors plan to support benchmarking in pytorch and tensorflow?

**Ethics:**

I am not an expert in clinical data, and I am unsure if the de-identification approach the authors used in Stanza's creation (the algorithm identified by Chambon et al. 2022) can guarantee the inability to reverse-engineer the patients' identities. That being said, the authors clearly indicate it is going through IRB approval and have an extensive ethics section; I would imagine that inclusion of this dataset is contingent on IRB approval.

One minor note - do the authors of Zhang et al. (2021) need to be notified of this?


**Limitations:**

It appears that the datasets are generally focused on chest anatomy. This is not a limitation of the author's efforts, but of the field. The authors can underscore this point by providing a meta analysis of the benchmark's contents.


**Opportunities For Improvement:**

My overall summary and contributions covers my rationale, but to reiterate:

- Metadata statistics on the following datasets can be extremely helpful in understanding current weaknesses and bias in the benchmark, and how to improve upon it.

- Benchmark evaluation of models that are not fine-tuned, if possible.


**Relation To Prior Work:**

Authors do a sufficient job illustrating (1) the other radiology NLP datasets (2) a need in the field for a universal benchmark to evaluate performance objectively across models and (3) the framing of tasks in the radiology space.

**Summary And Contributions:**

This paper proposes a Radiology NLP benchmark to assess language models' proficiencies toward both (1) natural language understanding (NLU) and (2) natural language generation (NLG). This benchmark, dubbed 'RaLEs', has 6 datasets with 4 different tasks including NER, RE, classification, and summarization.

There are several key areas the authors provide as novel contributions:

- A combined benchmark explicitly focused on radiology (text) data. While radiology datasets exist in the space, a curated effort in a benchmarking suite does not exist to compare and estimate radiology models. Many previously existing models are trained or evaluated on private datasets, due to confidentiality laws, making it difficult to compare performance. Within the benchmark is 1 novel dataset curated by the authors.

- A panel of model evaluations across 16 LLMs on this proposed benchmark, fine-tuned to adapt to some novel tasks (i.e. summarization).

Given there are no objective evaluation strategies to compare radiology model performance, I am inclined to accept this paper as it answers an unmet need in the space (there is a minor caveat in that some of the datasets will require external approval ex: MIMIC, however this is to ensure appropriate data use and I think this is acceptable).

There are a few comments that would improve the contents of the paper. They are as follows:

**Greater granularity on dataset statistics**

The authors collect 5 existing datasets and curate 1 new one. One concern is that a majority of these reports are focused on chest x-rays or CT. Future studies may want to consider building new datasets with orthogonal information to supplement what currently exists. Can the authors quantify meta statistics on what fraction of the data is split by:

* Anatomy (the authors use Chest or "All"; can you elaborate on what "all" encompasses and provide some statistics of what percent of the total data is about what anatomy? For example, Section 3 indicates head, chest, abdomen, spine, and sinuses are all possible options for 'all')

* Modality (XR, CT, MRI): according to table 1, many of the datasets are only 1 modality, but the last dataset (BioNLP) has both CT and MRI data; would the authors provide aggregate statistics and individual statistics? This can help future evaluations stratify their data to understand failure modes.

* Average length of reports between each data source: the authors even note that XR data may be more succinct than CT or other modalities. It would be good to quantify these differences

* Duplication/overlapping information: BioNLP draws from MIMIC-III which is also present in the benchmark; MEDIQA 2021 draws from MIMIC-CXR; is there any concern that some examples might be duplicated from these datasets curation? Penalizing or rewarding models in a benchmark for the same example due to aggregated sources would be an unfair metric.

* Data Splits: When creating train/val/test splits (ex: RadSpRL), can the authors provide some background information on how these splits are determined? Are there any deliberate attempts to stratify against bias? The MIMIC-III procedure is identified, but I am unclear if RadSpRL came with pre-made splits or not).

* Is there a reason the authors only selected for CT reports in MIMIC-III? I understand that this is to encourage diversity of reports, but there are nearly 500K reports (as mentioned in the appendix) that contain radiology terms. Moreover, this biases the NLU aggregate benchmark to be heavily concentrated on CT, due to the enormous size of MIMIC-III compared to the other datasets.

Simple helper functions, or a table to annotate some of these meta statistics in the appendix can be vastly beneficial to understand where current limitations lie, and how to build upon this benchmark in future iterations or by other collaborators.

**Model Evaluation additions**

In section 4.2.3, the authors must adapt some models provided to perform the summarization task. The authors "construct an encoder-decoder model for sequence generation using the same pretrained model as the encoder and decoder, and learn a cross-attention layer". Can the authors elaborate whether there is any freezing performed on layers? Section B elaborates fine tuning v. freezing with a grid hyperparameter search but appears to be with regards to the RE tasks.

The results of Table 3 are interesting, suggesting that post fine-tuning, many of the models are extremely comparable. This is curious, as one may imagine domain specific models should be (ideally significantly) more performant than baseline models. Inspection of the table indicates that all models are very close in performance, suggesting that fine-tuning can make any model (regardless its pre-trained prior) viable.

Given some of these models that are specifically trained on clinical/biological/even radiology data use very large pre-training datasets, it's interesting to see that even a much smaller fine-tune dataset confers good performance across a variety of different datasets. This might open the question of (1) how valuable is a pre-trained base and (2) how difficult it is to learn the radiology dataset text representation. Fine-tuning all these models confounds the answer to these questions.

Will the authors include baselines from models without fine-tuning (i.e. BERT-base without fine tuning?). It would be informative to understand if base models differ; particularly between an uninformed pre-trained base (ex: BERT) versus a specialist (RadBERT).

Lastly, the authors indicate in section 2.1 that many pre-existing models in radiology NLP seem to favor classic ML or even rule-based heuristics. While the benchmark seems to support LLM evaluations, it may be useful to consider how one might report results on a non-LLM model (granted, such models are highly unlikely to perform all tasks in the benchmark, but may be able to perform some subset. Given the specialized nature of medical data, it may still be useful to know what the best-in-class is.). This is a minor note, and does not depreciate from the fact some benchmark of performance measure for LLMs should exist.

**Documentation is sparse and unfinished**

See the documentation section for further details; I understand this is a work-in-progress. However, I would expect this to be completed upon formal acceptance.

---

> ### Author Response · Authors · 2023-08-21
> **Thank you for your detailed review.**
>
> > Greater granularity on dataset statistics. Can the authors quantify meta statistics on what fraction of the data is split by: Anatomy, Modality
>
> We have added clarifications about what anatomies are encapsulated in “All”, modifying it as a “Varied” label, and clarifying that this includes Abdomen, Pelvis, Neck, Spine, Head, etc. Additionally, we have created additional dataset statistics, now available in Appendix A and B. These better illustrate the class imbalance as well as facilitate error analyses. To illustrate this, we characterize the performance of the best current procedure selection model in Appendix I, providing per-class performance metrics, and showing that the relationship between prevalence in the training/testing data and class performance is not strictly direct, as classes with low number of examples may be easier to classify than larger but less specific classes.
>
> >Average length of reports between each data source: the authors even note that XR data may be more succinct than CT or other modalities.
>
> We have created summary statistics for the average length of the reports. We have identified that MEDIQA2021 reports (chest x-ray reports are  63+/-31 tokens long (average +/- standard deviation). In contrast, BIONLP2023 reports (CT, MRI reports), are on average 159 +- 85 tokens. We have provided additional report length statistics in Table 13 in the revised version of the manuscript.
>
> >Duplication/overlapping information: BioNLP draws from MIMIC-III which is also present in the benchmark; MEDIQA 2021 draws from MIMIC-CXR; is there any concern that some examples might be duplicated from these datasets curation? Penalizing or rewarding models in a benchmark for the same example due to aggregated sources would be an unfair metric.
>
>
> The reports used for the BioNLP2023 evaluation in RaLEs correspond to CT and MR reports. In contrast, the MEDIQA2021 reports are all Chest X-ray reports. Therefore, there is no overlap in samples between these evaluation sets.
>
>
> >Data Splits: When creating train/val/test splits (ex: RadSpRL), can the authors provide some background information on how these splits are determined? Are there any deliberate attempts to stratify against bias? The MIMIC-III procedure is identified, but I am unclear if RadSpRL came with pre-made splits or not).
>
> The original RadSpRL authors did not provide splits, but rather performed evaluations with 10-fold cross validation [1]. In line with the evaluation setting proposed in the original publication, we created splits by randomly distributing reports with positive entity mentions across train/dev/test We did not perform cross-validation given the amount of models and other datasets evaluated in our benchmark. We also note that reports for this dataset are sourced from the Open-I dataset, in which each report is collected from a different patient.
>
> [1]Datta, S., Si, Y., Rodriguez, L., Shooshan, S. E., Demner-Fushman, D., & Roberts, K. (2020). Understanding spatial language in radiology: Representation framework, annotation, and spatial relation extraction from chest X-ray reports using deep learning. Journal of biomedical informatics, 108, 103473.
>
> > Is there a reason the authors only selected for CT reports in MIMIC-III?
>
> Our decision to focus exclusively on CT reports in MIMIC-III was motivated by our goal of developing a dataset centered on common protocol selection. CT stands out as the predominant volumetric imaging modality, with an annual count exceeding 80 million scans in the U.S.
> While we recognize the potential benefit of incorporating other imaging modalities like MRI, the primary constraint lies in the manual efforts required for mapping and standardizing procedures titles to standardized terminologies. Given this, our initial approach was to prioritize the most prevalent imaging modality.
> That said, we envision RaLEs as an evolving benchmark, open to contributions from our team and the broader research community. To facilitate this, we've outlined guidelines on our codebase ([link](https://github.com/StanfordMIMI/RaLEs/blob/main/datasets/README.md#adding-a-dataset-to-the-rales-benchmark)) detailing how researchers can integrate datasets into the RaLEs benchmark and recommended practices for dataset selection and model benchmarking.

---

> ### Author Response · Authors · 2023-08-21
> **Thank you for your detailed review (part 2/3 of response)**
>
> > helper functions, or a table to annotate some of these meta statistics in the appendix can be vastly beneficial to understand where current limitations
>
> In response to your suggestion, we've improved our codebase with tools designed to provide a deeper understanding of dataset characteristics. This includes scripts to generate statistics, such as report lengths (e.g. see [here](https://github.com/StanfordMIMI/RaLEs/blob/main/datasets/generate_dataset_length_statistics.py)). We’ve also included these in the manuscript as mentioned above.
> Moreover, we've extended the error analysis for both the procedure prediction and report summarization tasks. This updated analysis delves into per-class performance, highlights limitations of leading models, and underscores areas where evaluation methodologies for summaries can be enhanced. The results of this analysis are summarized in the newly introduced Appendix I and J.
> Finally, to further streamline future research, we intend to make the top-performing models for each class available on huggingface.
>
> >In section 4.2.3, the authors must adapt some models provided to perform the summarization task. [...] Can the authors elaborate whether there is any freezing performed on layers?
>
> In our approach we fine-tuned all the weights (i.e. encoder, decoder and cross-attention layers) as we found this to lead to improved performance in preliminary experiments. We have added this clarification to the manuscript.
>
> >Will the authors include baselines from models without fine-tuning (i.e. BERT-base without fine tuning?). It would be informative to understand if base models differ; particularly between an uninformed pre-trained base (ex: BERT) versus a specialist (RadBERT).
>
> Thank you for these points - these are interesting and important observations. In Table 11 of the manuscript, we have included evaluations for a linear-probe, in which the base model weights are maintained frozen, and only a linear classification head is fine-tuned. Given that these models cannot achieve zero-shot capabilities, we believe this is the closest evaluation to your request. The results suggest that embeddings generated by general models, such as BERTbase, are more linearly separable compared to RadBERT1/2 embeddings. Further, by providing evaluations with 1 and 10% training labels, summarized in Tables 9 and 10, we further illustrate the data efficiency of the different methods, currently favoring clinical-specific models trained from scratch such as GatorTron.
>
> >While the benchmark seems to support LLM evaluations, it may be useful to consider how one might report results on a non-LLM model (granted, such models are highly unlikely to perform all tasks in the benchmark, but may be able to perform some subset.
>
> We value your suggestion to consider non-LM in our evaluation. While RaLEs is designed with LM evaluations in mind, it remains flexible enough to assess models specialized in specific tasks. To exemplify:
>
> * In the updated manuscript, we've included evaluations for RadiologyGPT, an LLM specializing in chest x-ray report summarization. This model was evaluated across both chest x-ray and other modality summarizations.
> * We are actively evaluating RadAdapt models [1], known for their state-of-the-art performance in radiology report summarization. We aim to share these results in a forthcoming update.
>
> Although we aren't aware of publicly available pre-existing models addressing individual tasks in our evaluations, we've made it easier for the community to contribute. We've introduced a guide in our codebase detailing how one can submit evaluation results for any or all tasks within RaLEs (see our guidelines). This encourages and facilitates the community in benchmarking newer models on the tasks enumerated in RaLEs.
>
> [1] Dave Van Veen, Cara Van Uden, Maayane Attias, Anuj Pareek, Christian Bluethgen, Malgorzata Polacin, Wah Chiu, Jean-Benoit Delbrouck, Juan Zambrano Chaves, Curtis Langlotz, Akshay Chaudhari, and John Pauly. 2023. RadAdapt: Radiology Report Summarization via Lightweight Domain Adaptation of Large Language Models. In The 22nd Workshop on Biomedical Natural Language Processing and BioNLP Shared Tasks, pages 449–460, Toronto, Canada. Association for Computational Linguistics.
>
> >Minor note - the citation style is somewhat hard to parse in the text
>
> Thank you for this observation. We have updated the citation style throughout.

---

> ### Author Response · Authors · 2023-08-21
> **Thank you for your detailed review (part 3/3 of response)**
>
> > The repo still looks under construction; documentation seems quite sparse. I would appreciate if the authors would provide the following instructions:
>
> >* Clear instructions on how to compile the benchmark dataset (I see there are scripts but I don't recall them linked in the top level README)
> >* A tutorial on how to bring a model to test across different datasets in the benchmark
> >* Are there plans for a leaderboard to reveal performance across each category, and in general? What are the ways new users can update the leaderboard, and how frequently will this be updated?
> >* Plans for maintenance, improvement, or longevity of the package?
>
> Thank you for this observation. We have improved the documentation throughout our codebase, and have also verified the reproducibility of our code. Specifically, we have provided additional instructions in each README for how each submodule works, with example commands that users may leverage to access the desired functionality. Furthermore, we have created a more detailed overview of the framework in our [main README](https://github.com/stanfordmimi/rales#usage), to orient users with different needs to specific components of our code.
>
> Additionally, we have created a live leaderboard to facilitate ongoing review of results ([see here](https://ralesbenchmark.github.io/)). We have provided instructions for how users may submit new results to the leaderboard, as well as how frequently it will be updated [here](https://github.com/StanfordMIMI/RaLEs/blob/main/results/README.md#submitting-to-the-leaderboard). Specifically, we have created a mechanism where users can submit a pull request with relevant information (contact information, model name, description, training data, fine-tuning strategy, datasets evaluated, test results, and associated paper and code). Furthermore, to facilitate reproducibility, we will create a direct download link for users with appropriate Physionet credentialing. The link will be available [here](https://vilmedic.app/datasets/papers). We will reply to this post once the link is live, which we expect will be before the end of the discussion period.
>
>
> >* Also, do the authors plan to support benchmarking in pytorch and tensorflow?
>
> Given the focus of the research community on PyTorch and huggingface libraries, RaLEs does not currently support Tensorflow. However, we have made our code modular such that Tensorflow users may download the data, and submit results without requiring the use of our training scripts which rely on PyTorch. Note the upcoming release of Keras 3.0 allows multi-backend support, which can further streamline this process.
>
>
> > I am unsure if the de-identification approach the authors used in Stanza's creation can guarantee the inability to reverse-engineer the patients' identities.  One minor note - do the authors of Zhang et al. (2021) need to be notified of this?
>
> Thank you for the observation. In addition to automated hidden-in-plain-sight deidentification, we have manually reviewed each report to confirm there is no leakage of PHI. Further, we have notified the authors of Zhang et al (2021), and note there is partial overlap between the authors, in particular Curtis Langlotz.

---

> > ### Author Response · Authors · 2023-08-28
> > **Our dataset download link is live**
> >
> > We have updated the link for data download, which is now live.

---

### Official Review · Reviewer_cMGK · 2023-07-21
**New Radiology Benchmark and CT procedure selection dataset.**

**Rating:** 8
**Confidence:** 3
**Clarity:** The paper is clearly written.

**Strengths:**

- First radiology domain NLP benchmark
- Evaluation of benchmark on 16 models from various domains and model size, showing new insights.
- Show more specialized models tend to do better for specialized domain tasks.
- Release new procedure selection dataset, which has 96,819 CT documents.

**Additional Feedback:**

Nice work!

**Correctness:**

The claims seem correct.
The newly released procedure selection dataset had procedures manually normalized by an MD and a board-certified radiologist.

**Documentation:**

Good

**Ethics:**

A potential concern I had is that the Stanza radiology datasets was de-identified using the hidden-in-plain-sight de-identification algorithm (Chambon 2022b). I read the abstract of that paper, and it claimed this algorithm performed better than the human labelers for de-identification. I am not familiar with de-identification to know what is good enough for public release.

Other than that, ethical section saids all licensing and research use agreements have been approved.

**Limitations:**

the author discussed the limitations adequately.
- The evaluation results between different models are oftentimes very close. Providing variance, as suggested by the author, would be useful.

**Opportunities For Improvement:**

- In Figure 1, explain why for many models (BioLinkBERTbase, DeBERTaV3, ELECTRA, and RoBERTa), the base model does better than the large model.
- In Figure 3, the range of RaLEs NLU score is small and there is a rough linear correlation between GLUE/BLURB score and RaLEs NLU score, which detracts from the authors claim that an additional radiology-specific benchmark is necessary. It would be interesting to see RadBERT1/2 plotted in Figure 3. Particularly if that has good performance in RaLEs but relatively poor performance in GLUE and/or BLURB.



**Relation To Prior Work:**

This paper states a lack of prior benchmarks in the radiology domain. Builds on prior biomedical benchmark but for a specialized domain.

**Summary And Contributions:**

This paper presents RaLEs, a new radiology domain benchmark for natural language understanding and generation. There has been benchmark for the general domain, such as GLUE or SuperGLUE, and benchmarks for the biomedical domain, such as BLURB or BLUE. But there are no benchmarks for radiology, which is of high interest currently for research and applications.  The RaLES consists of 6 datasets and 4 tasks. The paper introduces a new procedure selection dataset and de-identifies and releases publicly the Stanza radiology dataset.

This paper provides comprehensive evaluation for all 6 datasets/tasks on 16 models from the general, biomedical, clinical, and radiology domains. The author states that <20% of prior work of NLP in radiology compared the proposed method with alternate approaches. So this study provides a great meta-analysis of model performance in this domain, showing models in clinical/radiology domain outperformed models in the general/biomedical domain.

---

> ### Author Response · Authors · 2023-08-21
>
> >In Figure 1, explain why for many models (BioLinkBERTbase, DeBERTaV3, ELECTRA, and RoBERTa), the base model does better than the large model.
>
> This is a good observation. We believe this to be the result of several compounding factors. First, this may stem from the nature of our evaluations. Popular benchmarks such as GLUE scores, which traditionally showcase the improved performance of the large model configurations, rely on sentence-level tasks, as opposed to several of our word or entity level classification tasks. Further, improvements seen by increasing model size are not seen across the board in other out-of-domain evaluations. For example, the [BLURB leaderboard](https://microsoft.github.io/BLURB/leaderboard.html) shows that PubMedBERT-base can outperform PubMedBERT large in some tasks. Similar patterns are observed in other domains [1]. Finally, as we show in Figure 2, improvements in alternate benchmarks do not translate into improvements in RaLEs.
>
> [1]Qiu, L., Shaw, P., Pasupat, P., Shi, T., Herzig, J., Pitler, E., ... & Toutanova, K. (2022). Evaluating the impact of model scale for compositional generalization in semantic parsing. arXiv preprint arXiv:2205.12253.
>
> > It would be interesting to see RadBERT1/2 plotted in Figure 3. Particularly if that has good performance in RaLEs but relatively poor performance in GLUE and/or BLURB.
>
> We have carried out the evaluation of RadBERT1/2 on BLURB and incorporated these new results Figure 3. As the reviewer hypothesized, RadBERT1/2 show improvements in RaLEs, but show worse performance in BLURB compared to other biomedical models. This helps clarify the non-linear relationship between RaLEs and existing biomedical benchmarks, such as BLURB. Further, it suggests there is room for domain-specific model adaptation that does not deter performance in other domains.
>
> >A potential concern I had is that the Stanza radiology datasets was de-identified using the hidden-in-plain-sight de-identification algorithm (Chambon 2022b). I read the abstract of that paper, and it claimed this algorithm performed better than the human labelers for de-identification. I am not familiar with de-identification to know what is good enough for public release.
>
> In addition to hidden-in-plain-sight we have manually verified the de-identification of each report for this dataset.

---

### Official Review · Reviewer_JDoA · 2023-07-23
**RaLEs**

**Rating:** 6
**Confidence:** 3

**Strengths:**

The authors make a convincing case that radiology reports can have systematic differences compared with other (even biomedical text), and that it would valuable to have a set of benchmarking tasks specific to radiology tasks. The 6 datasets (including 2 previously not accessible ones) across 4 tasks form a nice start for such a benchmarking set of datasets. Improvements to NLP for radiology-specific tasks could have be of positive downstream use for either research or patients hoping to make better sense of their radiology reports.

**Additional Feedback:**

N/A

**Clarity:**

The paper is generally well written, though one minor thing that would help clarity is to use brackets / parentheses for citations, as this completely in-line format disrupts the flow of the text.

**Correctness:**

In general, the evaluation methods look correct. There are some small inconsistencies in the code (e.g., in `evaluate_best_procedure.sh`, `evaluate_models.py` is referenced, but it doesn’t exist. This does just look like a mismatch in versioning since everything else references `evaluate_procedure_models.py` so that is likely the correct script).

**Documentation:**

Yes.

**Ethics:**

No.

**Limitations:**

One of the main potential limitations already mentioned above is that of potential data leakage (either by patient, radiologist, etc.) and should be addressed by the authors. The authors do acknowledge several important ethical and societal concerns (e.g., using models for clinical care or advice without tailored studies).

**Opportunities For Improvement:**

There is mention of how the radiology reports were split into train/dev/test splits at the report level for RadSpRL and the MIMIC-III procedure selection dataset, but in general, across the various datasets, how the splits were done is not always easily evident. This is critical because one recurring theme in many biomedical benchmarking studies is unintentional data leakage. For example, if RadSpRL has any patients that have multiple reports and dataset splits did not consider stratifying by patient, then this could be problematic.

It would be interesting to see some rationale for why the RadBERTs did not necessarily outperform the other models. After the authors made the strong case that radiology reports should be considered a different domain from general English text / biomedical / clinical, then it seems intuitive that models trained specifically on radiology reports would have strong performance.

In the dataset descriptions in 4.1, it is often not completely clear which annotations were part of the original dataset and which are novel contributions as part of this paper.

**Relation To Prior Work:**

Yes.

**Summary And Contributions:**

Chavez et al develop a new benchmarking dataset for radiology reports that they term RaLEs. They organize 6 datasets (including 1 newly curated one and 1 newly de-identified dataset) across 4 tasks (NER, RE, procedure classification, and summarization). They then benchmark 16 different pre-trained language models, including models trained on general non-domain-specific English text, biomedical text, clinical text, as well as radiology reports. The results of the benchmarking study show that no single pretrained model dominates all tasks, though almost none of the top performers are general purpose (non-domain-specific) language models.

---

> ### Author Response · Authors · 2023-08-21
> **Thank you for your detailed review.**
>
> >There is mention of how the radiology reports were split into train/dev/test splits at the report level for RadSpRL and the MIMIC-III procedure selection dataset, but in general, across the various datasets, how the splits were done is not always easily evident. This is critical because one recurring theme in many biomedical benchmarking studies is unintentional data leakage.
>
> >One of the main potential limitations already mentioned above is that of potential data leakage (either by patient, radiologist, etc.) and should be addressed by the authors.
>
> This is a relevant and important point. For RadSpRL, which leverages the Open-I dataset, each report is sourced from a different patient. Thus, patient leakage is not an issue.
>
> For the procedure selection dataset, we have identified that the data was not originally split at the patient level, and instead it was performed at the report level favoring a hierarchical split based on the labels. Based on your suggestion we have carried out sub-analyses for patients in the test set unseen during training or development. We observed decreased accuracy in this subset of patients across model types (on average a decrease of 16% accuracy compared to the full test set, new Figure 6). We hypothesize that in realistic application settings, institutions may wish to make inference both on previously seen as well as previously unseen patients. Therefore, it would be useful to know how a model will perform on a mix of the two, as well as on unseen only (giving an idea of generalizability to new patients). We now provide two test/dev sets so that model developers can now keep both scenarios in mind: seen and unseen patients. We will also provide document level identifiers that can be mapped to original MIMIC tables to facilitate exploration of other potential biases in the future.
>
> >It would be interesting to see some rationale for why the RadBERTs did not necessarily outperform the other models.
>
> We hypothesize this may be due to two limitations of existing approaches.
> * **Continued Pretraining Limitations**: Our primary hypothesis revolves around the potential shortcomings of the current model adaptation strategies. Specifically, further pretraining of pre-existing BERT models on radiology datasets might not be as effective as initially anticipated. Prior literature [1,2,3] suggests that training models from scratch on domain-specific datasets yields better results, provided the pre-training datasets are adequately large.
> * **Data Size for RadBERTs**: A secondary consideration pertains to the volume of data used for pretraining RadBERTs. Current implementations adapt their models with ~4 million radiology reports. This is roughly an order of magnitude less text than what BERT-base or PubMedBERT models are trained on. We suspect that this data disparity might be a significant factor in RadBERTs not reaching their anticipated performance levels.
> In essence, our findings underscore the necessity to re-evaluate the training strategy for domain-specific models like RadBERTs. They also emphasize the potential need for larger training datasets or more data-efficient adaptation techniques for radiology models.
>
> [1] Gu, Y., Tinn, R., Cheng, H., Lucas, M., Usuyama, N., Liu, X., ... & Poon, H. (2021). Domain-specific language model pretraining for biomedical natural language processing. ACM Transactions on Computing for Healthcare (HEALTH), 3(1), 1-23.
>
> [2]Beltagy, I., Lo, K., & Cohan, A. (2019). SciBERT: A pretrained language model for scientific text. arXiv preprint arXiv:1903.10676.
>
> [3]Sanchez, C., & Zhang, Z. (2022). The Effects of In-domain Corpus Size on pre-training BERT. arXiv preprint arXiv:2212.07914.
>
> >In the dataset descriptions in 4.1, it is often not completely clear which annotations were part of the original dataset and which are novel contributions as part of this paper.
>
> Thank you for highlighting this. We've now clarified in Section 4.1 which annotations are original and which are our contributions. Additionally, Table 1 has a new column indicating datasets newly released in this paper.
>
> >There are some small inconsistencies in the code (e.g., in evaluate_best_procedure.sh, evaluate_models.py is referenced, but it doesn’t exist. [...]).
>
> Thank you for noticing this! We have updated the code to ensure the reproducibility of the evaluation in an end-to-end fashion. Further, we have also greatly improved our documentation to facilitate further analyses.
> >The paper is generally well written, though one minor thing that would help clarity is to use brackets / parentheses for citations, as this completely in-line format disrupts the flow of the text.
>
> Thank you for the suggestion. We have modified the citation style to numbered/brackets instead of the previous in-line format.

---

> > ### Comment · Reviewer_JDoA · 2023-08-25
> >
> > It is nice to see that the authors have now also included a patient-stratified setup and updated the experiments accordingly, and also took into account the other items and adjusted accordingly. The authors also do a good job of addressing my other concerns and clarifying the contributions of this manuscript better. I've bumped my score up to reflect the improvements as well!

---

### Official Review · Reviewer_w7FQ · 2023-08-02
**A good paper**

**Rating:** 6
**Confidence:** 4
**Correctness:** None
**Clarity:** None

**Strengths:**

The strengths of the paper are:

- 1. The paper is well-written. It has a good summary of the situations in the radiology NLP area and existing relevant datasets.
- 2. One new dataset is created to complement existing datasets.
- 3. The authors benchmark various existing models designed for different domains to find out their performance differences on the benchmark. This better help the readers to understand the progress in the radiology NLP.

**Additional Feedback:**

None

**Documentation:**

None

**Limitations:**

No particular ones.

**Opportunities For Improvement:**

- 1. The scores on some datasets are high already (>90\%). Therefore, it'd be better to do some error analysis to better illustrate the datasets for the readers.

**Relation To Prior Work:**

None

**Summary And Contributions:**

In this work, the authors presented a benchmark for natural language understanding and generation in radiology. The benchmark includes both NLU and NLG tasks with five existing datasets and one newly created dataset. In addition, they compared the performance of models designed for different domains on all the datasets.

---

> ### Author Response · Authors · 2023-08-21
> **Thank you for your suggestions.**
>
> Thank you for your review. We have included additional error analyses for the best in-class models in the Appendix (Sections I, J, Tables 14-16, Figures 5,6). We have focused on the datasets with the greatest potential for performance improvement: procedure selection and report summarization.
> For **procedure selection**:
> * We observe that the performance for each class is partially explained by label availability. Rarer classes such as “CT abdomen with IV contrast” tend to have worse performance. Interestingly, this is not universally the case, as some underrepresented classes, such as “CT Guided Str loc–W contr IV”, exhibit higher performance. We hypothesize that certain classes with highly specific indications are easier to classify, even with smaller sample sizes in the training data.
> * Furthermore, as illustrated in Table 14, there are instances where the provided indication alone is insufficient to reliably determine the ground truth, exemplified by the indication “Please evaluate for interval progression of hemorrhage”. This observation suggests the opportunity to develop improved datasets in the future, possibly integrating clinical notes with the examination indications. This could allow for modeling approaches that can request additional information in case the provided one is insufficient.
> * Finally, we find that models may be confidently incorrect, as shown in the examples in Table 14 and further quantified by our calibration error statistics, such as ECE, listed in Table 7. Calibrating predictions may be another avenue for further research.
>
> For report summarization, we have listed in Table 16 various example reference and generated impressions across the two datasets. We observe that certain metrics can be on occasion contradictory for the same (generation, ground truth) pair. For example, when additional recommendations based on medical expertise are provided in the hypothesis, metrics such as ROUGE impose additional penalty on the prediction if these are not present in the reference. Conversely, in scenarios where no specific disease entities are mentioned in the ground truth, metrics like F1 RadGraph unfairly penalize the models given that they rely on the explicit mention of diseases and their relation to anatomies. This is also affected by the lack of validation of automated RadGraph annotation on non-Chest X-ray reports, as well as lack of normalization to a standardized nomenclature for entities. Such examples illustrate the need for refining summarization evaluation metrics. They also highlight the importance of developing models that not only understand the immediate report but can also draw upon broader context, enabling them to offer recommendations rooted in both the findings and best clinical practice.

---

### Author Response · Authors · 2023-08-28
**Thank you for your feedback, we’ve updated our manuscript!**

We thank the reviewers for their valuable feedback. We are deeply appreciative of their acknowledgment of our contributions to the radiology NLP domain. Their feedback highlights the significance of our comprehensive overview in the field (Reviewer w7FQ) and the relevance of benchmarking tailored for radiology reports (Reviewer JDoA).

Our thorough evaluation of over 16 models enables RaLEs to offer the community vital insights (as highlighted by Reviewer cMGK). The reviewers' appreciation of our emphasis on state-of-the-art models and reproducibility, manifested through a data and model fine-tuning standardization framework, is especially encouraging (Reviewer mpiG; Reviewer SKzi).

Based on the comments received, we have uploaded a revised version of our paper. The main changes to the updated manuscript include:
* Updated table and figure quality in the main manuscript (recommended by Reviewers JDoA, mpiG)
* Expanded error analysis, introducing Appendix I, J, as recommended by Reviewers w7FQ, SKzi)
* Included evaluations of new methods (RadiologyGPT, as recommended by Reviewer mpiG, as well as RadAdapt)
* Updated citation style to facilitate readability (recommended by Reviewers JDoA, SKzi).

We’ve also improved our documentation throughout our codebase (as recommended by Reviewers JDoA, mpiG), including instructions for how to use it, how to submit results to the leaderboard, and how to submit new datasets. Further, we’ve created a website featuring our leaderboard, available at https://ralesbenchmark.github.io. In addition, we’ve now created a direct download link at https://vilmedic.app/datasets/papers, to make it easier for users with appropriate credentials to access our processed data, based on suggestions from Reviewers mpiG and SKzi.

---

### Decision · Program_Chairs · 2023-09-22

**Decision:**

Accept (Poster)

**Comment:**

As highlighted by the reviewers, this strength of this work lies in the value of the combined benchmark including existing and new datasets, and the fact that it is specific to the radiology field [w7FQ, cmGK, SKzi, mpiG]. Reviewers also highlighted the value of the empirical study of models [w8FQ, cmGK, mpiG] and the code framework [SKzi, mpiG].

Various weaknesses were identified by reviewers, however in almost all cases I note that the authors addressed the questions and (in my consideration) resolved them, however the reviewers did not adjust their scores in all but one case, hence I consider the current aggregate score a slight under-estimate. Some weaknesses which were addressed were: providing further detail on splits [JDoA], interpreting particular findings [cmGK], manually confirming that de-identification was successful for the Stanza dataset [cmGK, mpiG], and adding a linear probe analysis for general-domain models [SKzi]. I note that one reviewer raised a concern that relatively little *new* data is provided in this benchmark [mpiG], however I tend to agree with the authors that this track welcomes useful collections of existing datasets; further the inclusion of the procedure mapping dataset on top of MIMIC-III and the release of the Stanza dataset is welcome.

A particular note on splits: the instructions for obtaining the datasets described on the code repository does not seem to be sufficient to obtain the correct splits, which is critical for a benchmark (as noted, RadSpRL does not come with canonical splits and hence the authors must additionally supply their split information). The dataset available for download (from this link: https://vilmedic.app/datasets/papers; I note the link on the README is to https://vilmedic.app/datasets/text which doesn't seem to work) includes pre-split data, as desired. Hence, I'm happy that the splits *can* be made available, however to ensure users of the benchmark follow the desired splits, it should be either be made clear that the only route to accessing the dataset is via said download link (in which case Physionet credentials are currently required to access any of the data), or to make the split information independently available such that it can be combined as needed.

A final note, it would be of likely interest to the community to evaluate recent large language models on these tasks (e.g. GPT-3.5, LLaMa-2) given such models could theoretically attempt these tasks with zero or few shot instruction.